# SPRING licenses S1P-mediated cleavage of SREBP2 by displacing an inhibitory pro-domain

Sebastian Hendrix [1], Vincent Dartigue [2], Hailee Hall[2], Shrankhla Bawaria [2], Jenina Kingma [1], Bilkish Bajaj[2], Noam Zelcer [1] ✉ & Daniel L. Kober [2] ✉

Site-one protease (S1P) conducts the first of two cleavage events in the Golgi to activate Sterol regulatory element binding proteins (SREBPs) and upregulate lipogenic transcription. S1P is also required for a wide array of additional signaling pathways. A zymogen serine protease, S1P matures through autoproteolysis of two pro-domains, with one cleavage event in the endoplasmic reticulum (ER) and the other in the Golgi. We recently identified the SREBP regulating gene, (SPRING), which enhances S1P maturation and is necessary for SREBP signaling. Here, we report the cryo-EM structures of S1P and S1P-SPRING at sub-2.5 Å resolution. SPRING activates S1P by dislodging its inhibitory pro-domain and stabilizing intra-domain contacts. Functionally, SPRING licenses S1P to cleave its cognate substrate, SREBP2. Our findings reveal an activation mechanism for S1P and provide insights into how spatial control of S1P activity underpins cholesterol homeostasis.

Site-one protease (S1P, also known as Membrane-bound transcription factor site-1 protease (MBTPS1), or Subtilisin kexin isozyme 1 (SKI-1)), is a membrane-bound protease that mediates the proteolytic activation of transcription factors, hormones, and enzymes required for lipogenesis, the endoplasmic reticulum (ER) stress response, and lysosomal biogenesis, among others[1,2]. The best-studied role for S1P is in cholesterol homeostasis, where it conducts the first of two cleavage steps that mature the transcription factors Sterol regulatory element binding proteins (SREBPs) to upregulate cholesterol uptake and biosynthesis[3,4]. Genetic deletion of S1P is embryonically lethal in mice[5] and zebrafish[6], and the deletion of S1P in mouse livers results in decreased cholesterol and fatty acid biosynthesis and decreased plasma cholesterol[5]. In humans, hypomorphic variants in *MBTPS1* have been linked to skeletal dysplasia and elevated lysosomal enzymes in the blood[7,8]. Additionally, S1P is exploited by Arena viruses to mature their viral glycoproteins[9–11] and by Hepatitis C virus (HCV), where S1P and SREBP are required to regulate the viral lifecycle[12]. Based on mutational analysis, the S1P cleavage motif is often described as RXXK↓ or RXXL↓. Mutation of the P4 Arg invariably eliminates cleavage by S1P and mutating the P1 residue to

Ala also inhibits proteolysis of certain S1P substrates[13–15]. A more complete description of the motif is also reported as RX(L/I/V)Z, where X is residues other than Pro and Cys, and Z is preferably Lys or Leu and is not Val, Pro, Cys, or Glu[1]. S1P substrate preferences are distinct among the subtilisin-like proteases of the mammalian secretory pathway and the structural basis for the specificity of S1P activity is not known.

Beginning from its N-terminus, S1P contains a signal peptide that mediates insertion into the ER lumen. The signal peptide is followed by two N-terminal pro-domains, termed the A-domain and B-domain, which chaperone the folding of S1P[1]. The pro-domains are followed by the Peptidase S8 protease domain, which harbors the catalytic triad consisting of D218, H249, and S414, along with an oxyanion hole residue, N338[3,16]. S1P terminates with a transmembrane helix followed by a short cytoplasmic tail (Fig. 1a). The catalytic reaction of subtilisin serine proteases is well-understood (reviewed in ref. 17). Briefly, the serine carries out a nucleophilic attack on the carbonyl group of the scissile peptide. This reaction is supported by the catalytic histidine's role as a general acid-base, mediating the transfer of a proton from the serine to the leaving amino group. The histidine is supported by the

¹Department of Medical Biochemistry, Amsterdam UMC, Amsterdam Cardiovascular Sciences and Gastroenterology and Metabolism, University of Amsterdam, Meibergdreef 9, 1105AZ Amsterdam, the Netherlands. ²Department of Biochemistry, The University of Texas Southwestern Medical Center, Dallas, TX 75390, USA. ✉e-mail: n.zelcer@amsterdamumc.nl; daniel.kober@utsouthwestern.edu

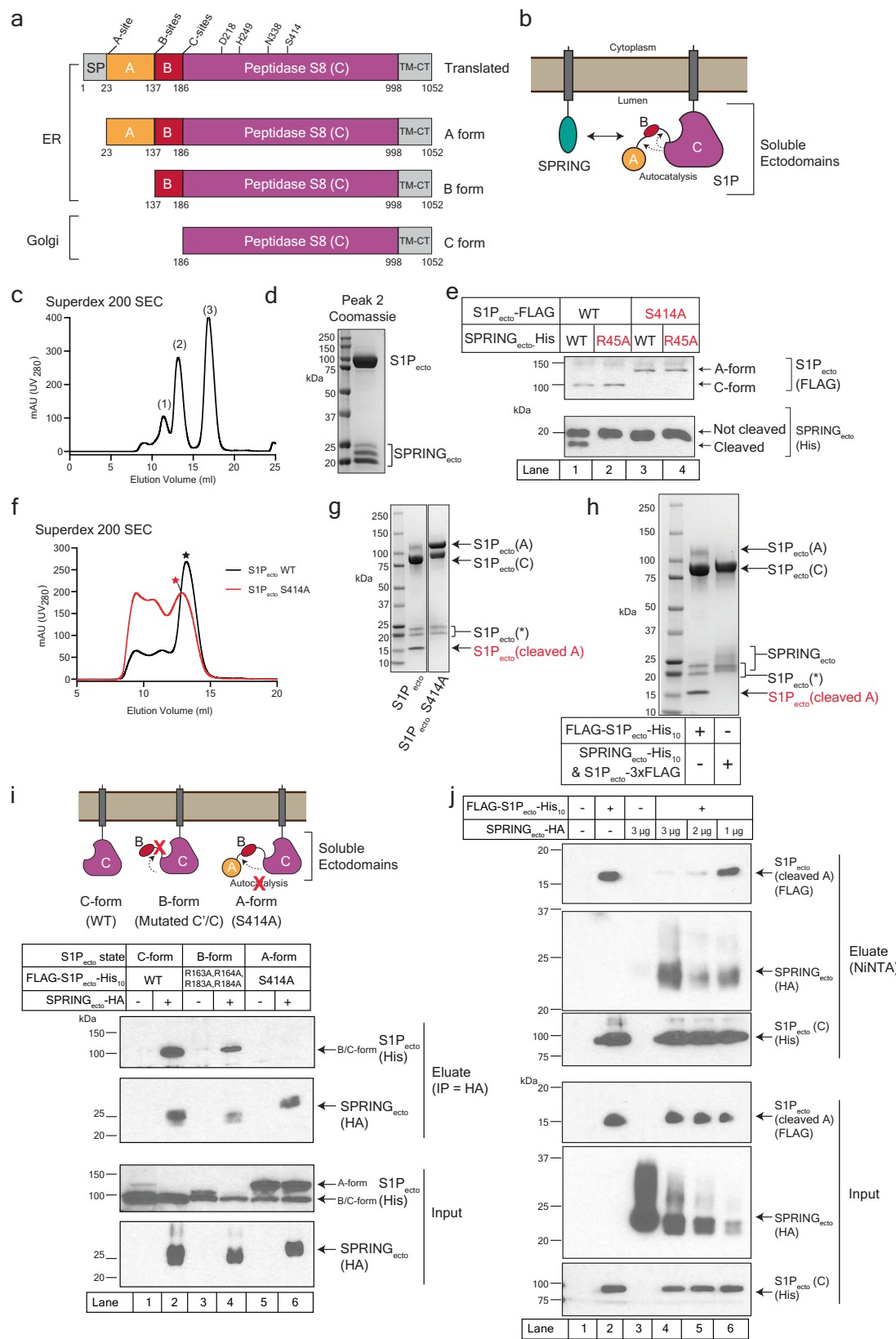

catalytic aspartate. The rate-limiting step is the formation of a tetrahedral oxyanion intermediate of the serine and substrate peptide. This intermediate is stabilized by donation of a hydrogen bond from the oxyanion hole residue. In subtilisin proteases, this residue is typically an asparagine, and mutation of this residue greatly decreases the activity of these proteases[18].

S1P matures through the sequential autoproteolysis of its pro-domains[19–21]. Removal of the signal peptide generates the A-form of S1P. This first cleavage site is also referred to as the A site. Between the A-pro-domain and B-pro-domain are two cleavage sites referred to as the B' and B sites. Autoproteolysis at the B' and/or B sites separates the A-domain and creates the B-form of S1P, and mutation of the B'/B sites

**Fig. 1 | SPRING$_{ecto}$ forms a complex with S1P$_{ecto}$. a** Domain organization of S1P and its autocatalytic processing. SP signal peptide; A = A-domain; B = B-domain; C = C-domain; TM-CT = Transmembrane-cytoplasmic domain. **b** Schematic for the S1P$_{ecto}$-SPRING$_{ecto}$ interaction through their soluble ectodomains and S1P auto-catalysis. **c** Purification of S1P$_{ecto}$-3xFLAG and SPRING$_{ecto}$-His$_{10}$ using superdex 200 gel filtration chromatography. **d** Coomassie-stained SDS-PAGE analysis of the S1P$_{ecto}$-SPRING$_{ecto}$ complex from Peak 2 in (**c**). **e** Cleavage of SPRING$_{ecto}$ by S1P$_{ecto}$. HEK 293 T cells were transfected with the indicated constructs and incubated for 72 h. Supernatants were harvested and subjected to PNGaseF treatment as described in Methods. Samples were subjected to immunoblot analysis using anti-His antibodies for SPRING$_{ecto}$-His$_{10}$ and anti-FLAG antibodies for S1P$_{ecto}$-3xFLAG. **f** Purification of FLAG-S1P$_{ecto}$-His$_{10}$ and FLAG-S1PS$_{ecto}$$^{S414A}$-His$_{10}$ using superdex 200 gel filtration chromatography. Stars indicate fractions collected for analysis by SDS-PAGE. **g** Coomassie-stained SDS-PAGE analysis of S1P$_{ecto}$ and S1P$_{ecto}$$^{S414A}$ from the fractions labeled with stars in (**f**). **h** Purified S1P$_{ecto}$ and S1P$_{ecto}$-SPRING$_{ecto}$ complexes analyzed by Coomassie-stained SDS-PAGE. Arrows or brackets indicate bands for S1P$_{ecto}$ and SPRING$_{ecto}$. The S1P$_{ecto}$ bands labeled with S1P (*) reflect non-specific degradation. **i** Co-IP of different forms of S1P$_{ecto}$ by SPRING$_{ecto}$. HEK 293 T cells were transfected with the indicated plasmids. After 72 h, supernatants were harvested and subjected to anti-HA pulldown as described in *Methods*. SPRING$_{ecto}$ was detected using anti-HA antibodies and S1P$_{ecto}$ was detected using anti-His antibodies. **j** Competitive pulldown of SPRING$_{ecto}$ and A-domain by S1P$_{ecto}$ C-domain. HEK 293 T cells were transfected with the indicated plasmids. After 72 h, supernatants were harvested and subjected to pulldown with NiNTA resin as described in Methods. SPRING$_{ecto}$ was detected using anti-HA antibodies. C-domain S1P$_{ecto}$ was detected using anti-His antibodies and A-domain S1P$_{ecto}$ was detected using anti-FLAG antibodies. All immunoblots are representative of at least 3 independent experiments. Source data are provided as a Source Data file.

has been shown to abolish maturation of S1P[20]. Between the B pro-domain and C domain are two sites referred to as the C′ and C sites. Autoproteolysis at the C and/or C′ site separates the B-domain and creates the C-form of S1P. Both the B-form and C-form of S1P are enzymatically active and can cleave SREBP in cells[20,22]. The B-form is generated in the ER but is rapidly transported to the Golgi where it matures to the C-form[20,22,23]. How the B-form to C-form transition is regulated is not well understood.

Spatial control of S1P activity underpins cholesterol metabolism in mammalian cells[24]. SREBPs are produced as ER-resident membrane proteins, where they interact with a cholesterol-sensing protein called Scap[25]. When the ER membranes of the cell are replete with cholesterol, Scap retains SREBP in the ER, and SREBP is not cleaved[26]. When ER membrane cholesterol levels decrease, Scap traffics to the Golgi in complex with SREBP[27]. In the Golgi, SREBPs are sequentially cleaved at two positions, first by S1P and then by Site-2 protease (S2P, encoded by *MBTPS2*)[15,28]. S2P cannot cleave SREBP when the cleavage of SREBP by S1P is inhibited[3,15,29–31]. Control of S1P activity and SREBP localization involves an elaborate regulatory system that is still not fully understood[24,32,33]. Critically, S1P must be prevented from cleaving SREBP precursors in the ER, where both the SREBP and S1P precursors are expressed.

Using a whole-genome functional genetic screen, we recently identified a previously uncharacterized protein, C12ORF49 (hereafter SREBP regulating gene, SPRING) that governs SREBP signaling[34,35], a finding subsequently confirmed independently by three other groups[36–38]. SPRING enhances the maturation of S1P from its precursor A-form in the ER to the active C-form in the Golgi[36,37,39]. Interestingly, treating cells with brefeldin A causes unregulated SREBP cleavage due to the relocation of the Golgi-resident S1P and S2P enzymes to the ER[40]. However, brefeldin A does not rescue SREBP cleavage in SPRING-deficient cells[36,37], suggesting that SPRING is required for S1P activity. As with S1P, global deletion of *Spring* is embryonically lethal in mice and inducible global deletion of *Spring* in adult mice also results in lethargy[35]. Liver-specific ablation of *Spring* in mice results in decreased hepatic SREBP signaling, decreased cholesterol and fatty acid synthesis, and a marked decrease in plasma cholesterol[41]. In humans, a *SPRING* coding variant is associated with HDL cholesterol and ApoA1 levels[41]. Yet despite its central role in SREBP signaling, why SPRING is functionally required for S1P activity remains unknown.

Previous reports characterizing exogenous, secreted S1P demonstrated that S1P interacts with its inhibitory pro-domain following proteolytic maturation[20,22,42]. This protease/pro-domain interaction evokes PCSK9, a serine protease from the same family that undergoes autocatalysis but remains enzymatically inactive because it cannot release its pro-domain[43–45]. The mechanism underlying the release of the inhibitory pro-domain from S1P has remained an enigma.

Here, we demonstrate a direct interaction between the ectodomains of SPRING and S1P and report the cryo-EM structures of the ectodomains of S1P and S1P-SPRING determined to sub-2.5 Å resolution. SPRING binds S1P competitively with the inhibitory S1P A-domain and thereby displaces this domain from matured S1P. Biochemical experiments show SPRING licenses S1P to cleave peptides derived from its canonical substrate, SREBP2, and cell-based experiments confirm the generalizability of our structural studies to the full-length proteins. Our findings shed light on a long-standing question pertaining to S1P activation and provide mechanistic insight into the role of SPRING in SREBP signaling.

## Results

### SPRING binding to S1P displaces the A-domain of S1P

To investigate the interaction between SPRING and S1P, we generated constructs expressing the soluble ectodomains of these proteins (Fig. 1b and Supplementary Fig. 1a). These ectodomains were expressed as secreted, soluble proteins from HEK 293 T cells for functional assays or else from HEK 293 s GnTI⁻ cells for large-scale purifications. HEK 293 s GnTI⁻ cells lack N-acetylglucosaminyltransferase I and are unable to produce complex N-linked glycans, which can be advantageous for structural studies[46]. For clarity, secreted proteins are labeled with the subscript "ecto." To test whether SPRING$_{ecto}$ directly interacted with S1P$_{ecto}$, we co-expressed S1P$_{ecto}$−3xFLAG and SPRING$_{ecto}$-His$_{10}$ in HEK 293 s GnTI⁻ cells grown in suspension (Supplementary Fig. 1a). From the supernatants we purified a complex containing stoichiometric amounts of S1P$_{ecto}$ and SPRING$_{ecto}$ using NiNTA affinity chromatography against the SPRING$_{ecto}$-His$_{10}$ followed by gel filtration chromatography (Fig. 1c, d peak 2, and Supplementary Fig. 1b). Excess SPRING$_{ecto}$ eluted largely as a monomer (peak 3), although it showed some oligomerization (peak 1) (Fig. 1c and Supplementary Fig. 1b). SPRING$_{ecto}$ contains 15 cysteines, creating the possibility of inter-molecular disulfide bonds. However, SPRING$_{ecto}$ from all gel filtration peaks migrated identically on SDS-PAGE under reducing or non-reducing conditions, and SPRING$_{ecto}$ could be modified with maleimide 5K-PEG, confirming the presence of at least one unpaired cysteine (Supplementary Fig. 1c). SPRING$_{ecto}$ co-expressed with S1P$_{ecto}$ from HEK 293 s GnTI⁻ cells is detected as three bands (Fig. 1d). Digesting this SPRING$_{ecto}$ with PNGaseF to remove N-linked glycosylations collapses SPRING$_{ecto}$ to two bands (Supplementary Fig. 1d). SPRING$_{ecto}$ contains an S1P cleavage motif near its N-terminus with the P4 Arg at residue 45[39] (Supplementary Fig. 1e). To test whether secreted SPRING$_{ecto}$ was cleaved by S1P$_{ecto}$, we co-expressed these proteins from HEK 293 T cells and treated the supernatants with PNGaseF to remove N-linked glycosylations. As expected, co-expression of S1P$_{ecto}$−3xFLAG and SPRING$_{ecto}$-His$_{10}$ results in two bands for SPRING$_{ecto}$ (Fig. 1e, lane 1). Co-expressing a P4-mutant SPRING$_{ecto}$$^{R45A}$ with S1P$_{ecto}$ or co-expressing a catalytically-inert S1P$_{ecto}$$^{S414A}$ with either form of SPRING$_{ecto}$ resulted in a single band for SPRING (Fig. 1e, lanes 2-4). These observations indicate SPRING$_{ecto}$ itself is a substrate for S1P$_{ecto}$[39]. Altogether, our biochemical analyses suggest the three SPRING$_{ecto}$ bands correspond to a mixture of gly-cosylated, non-glycosylated, and cleaved SPRING$_{ecto}$.

Next, we purified soluble S1P$_{ecto}$ without SPRING$_{ecto}$. For these experiments, we utilized a FLAG-S1P$_{ecto}$-His$_{10}$ construct, which was purified as a secreted product from conditioned HEK 293 s GnTI⁻ supernatants. We also purified FLAG-S1P$_{ecto}$^{S414A}-His$_{10}$, which lacks enzymatic activity[20]. Both S1P$_{ecto}$ and S1P$_{ecto}$^{S414A} could be purified to homogeneity using NiNTA affinity and gel filtration chromatography (Fig. 1f, g). Because the full-length version of S1P^{S414A}, which cannot mature, has been previously shown reside in the ER[23], we suspect the secretion of S1P$_{ecto}$^{S414A} is an artifact of the strong over-expression regime in HEK cells. Previous reports have shown the cleaved A-domain can be co-purified with S1P$_{ecto}$[20,22], and in our hands the cleaved A-domain of S1P$_{ecto}$ also co-purifies with S1P$_{ecto}$ (Fig. 1g). This cleaved band is not produced by the inactive S1P$_{ecto}$^{S414A} (Fig. 1g). We also observed a doublet band in both S1P$_{ecto}$ and S1P$_{ecto}$^{S414A} (Fig. 1g). It is possible that this doublet is the result of non-specific degradation in the S1P$_{ecto}$ B-domain as the B-domain peptide is unstructured and likely prone to proteolysis[22,47]. Another possibility is that the exogenous S1P$_{ecto}$ is cleaved at its C-sites in trans by endogenous S1P, as previously reported[20]. We have not explored this doublet band further.

FLAG-S1P$_{ecto}$-His$_{10}$ purified using NiNTA chromatography (without SPRING$_{ecto}$) and S1P$_{ecto}$−3xFLAG purified using NiNTA chromatography through its interaction with SPRING$_{ecto}$-His$_{10}$ were compared using Coomassie-stained SDS-PAGE (Fig. 1h). While the FLAG-S1P$_{ecto}$-His$_{10}$ sample contains some immature A-form S1P and the cleaved A-domain, S1P$_{ecto}$−3xFLAG purified through SPRING$_{ecto}$-His$_{10}$ lacks both of those bands as well as the non-specific degradation bands. Additional biophysical analysis of the SPRING$_{ecto}$-S1P$_{ecto}$ complex using mass photometry (MP) (Supplementary Fig. 1f) and size exclusion chromatography with multi-angle light scattering (SEC-MALS) (Supplementary Fig. 1g) showed that the complex has a solution MW of 116−120 kDa, a very close match to the expected MW for a heterodimeric complex of C-form S1P$_{ecto}$ and SPRING$_{ecto}$ (114.4 kDa without glycosylation) and smaller than the masses for a complex containing the A or the A and B-domains of S1P$_{ecto}$ (Supplementary Fig. 1h). We have been unable to detect whether the cleaved B domain is bound to either S1P$_{ecto}$ or SPRING$_{ecto}$-S1P$_{ecto}$ owing to its small size (~3-5 kDa depending on the combination of B′/B and C′/C cleavage sites).

Based on these initial results, we designed an experiment to test whether SPRING$_{ecto}$ binds a specific form of S1P$_{ecto}$. We generated three forms of S1P$_{ecto}$ in the FLAG-S1P$_{ecto}$-His$_{10}$ background: 1) a WT S1P$_{ecto}$ that undergoes the normal cleavage to the C-form; 2) S1P$_{ecto}$ with the C′ and C sites mutated (R163A, R164A, R183A, and R184A) to allow cleavage of the A pro-domain but not the B pro-domain, referred to as B-form S1P$_{ecto}$; and 3) S1P$_{ecto}$ harboring the catalytic S414A mutation that traps this S1P$_{ecto}$ in the A-form (Fig. 1i, top). We co-expressed the FLAG-S1P$_{ecto}$-His$_{10}$ proteins alone or with SPRING$_{ecto}$ containing a C-terminal HA epitope tag (Supplementary Fig. 1a). The input samples show the mutant S1P proteins in the precursor stages expected from the mutations, with S1P$_{ecto}$^{S414A} largely in the A-form and the WT and B-form constructs in their matured forms. Note there is some degradation of the inactive S1P$_{ecto}$^{S414A} that is reminiscent of the purified protein sample in Fig. 1g. Anti-HA resin was used to immunoprecipitate (IP) SPRING$_{ecto}$-HA, and eluates were probed with anti-HA and anti-His antibodies to detect SPRING$_{ecto}$ and S1P$_{ecto}$, respectively. SPRING$_{ecto}$ could co-immunoprecipitate (co-IP) the C-form and B-forms of S1P$_{ecto}$ (Fig. 1i, lanes 2 and 4). In contrast, an interaction between SPRING$_{ecto}$ and A-form S1P$_{ecto}$ was not detected (Fig. 1i, lane 6). Note, the SPRING$_{ecto}$ that was co-secreted with S1P$_{ecto}$^{S414A} (lane 6) migrated at a slightly higher size on SDS-PAGE compared to the SPRING secreted with WT or B-form S1P (lanes 2 and 4), consistent with the inability of S1P$_{ecto}$^{S414A} to cleave SPRING$_{ecto}$.

This apparent competition between the A-domain of S1P$_{ecto}$ and SPRING$_{ecto}$ was also observed in a reciprocal experiment in which we pulled down S1P$_{ecto}$ instead of SPRING$_{ecto}$. We transfected HEK 293 T cells with FLAG-S1P$_{ecto}$-His$_{10}$ and varying amounts of SPRING$_{ecto}$-

HA plasmid. Conditioned supernatants were harvested and the C-domain of S1P$_{ecto}$ was precipitated using NiNTA resin. In the absence of co-transfected SPRING$_{ecto}$, the cleaved A-domain of S1P$_{ecto}$ co-eluted with the C-form of S1P$_{ecto}$ (Fig. 1j, lane 2). SPRING$_{ecto}$-HA was present in the eluate only when co-expressed with FLAG-S1P$_{ecto}$-His$_{10}$, confirming their interaction in this format (Fig. 1j, lanes 3-6). Notably, the amount of cleaved, FLAG-tagged S1P$_{ecto}$ A-domain in the NiNTA eluate was inversely related to the amount of transfected SPRING$_{ecto}$-HA cDNA, further supporting the notion that the A-domain of S1P$_{ecto}$ and SPRING$_{ecto}$ competitively bind the C-domain of S1P$_{ecto}$.

## SPRING enables efficient cleavage of an SREBP2-derived substrate by S1P

To test whether the interaction with SPRING$_{ecto}$ affects the enzymatic activity of S1P$_{ecto}$, we designed a peptide containing 12 amino acids from SREBP2 (residues 516-527) that includes the S1P cleavage motif[15,28]. This peptide contains an N-terminal Methyl Cumaryl Amide (MCA) fluorophore that is quenched by a C-terminal 2, 4-dinitrophenyl (DNP) group when the peptide is intact, but which fluoresces when the peptide is cleaved and the fluorophore de-quenches (Fig. 2a). Similar S1P activity assays have been reported previously[21,23,42,47]. We observed robust cleavage of the SREBP2 peptide using the SPRING$_{ecto}$-His$_{10}$/ S1P$_{ecto}$−3xFLAG complex that could be inhibited with the S1P inhibitor PF-429242[39]. In contrast, we did not observe protease activity with FLAG-S1P$_{ecto}$-His$_{10}$ purified without SPRING$_{ecto}$, the catalytically "dead" FLAG-S1P$_{ecto}$^{S414A}-His$_{10}$, or SPRING$_{ecto}$-His$_{10}$ (Fig. 2b). Moreover, no activity was detected using a peptide where the P4 Arg was changed to Ala, consistent with cell-based mutagenesis experiments[39,48] (Fig. 2c). These controls show the SREBP2-cleaving activity in the S1P$_{ecto}$-SPRING$_{ecto}$ sample is not likely to be a contaminating protease. To exclude the possibility that the N-terminal FLAG tag on our FLAG-S1P$_{ecto}$-His$_{10}$ construct was responsible for the lack of S1P activity when S1P was purified without SPRING, we also purified S1P$_{ecto}$−3xFLAG using anti-FLAG affinity resin and gel filtration chromatography (Supplementary Fig. 1a, i). Real-time kinetic measurements with this sample also showed no detectable activity with this S1P$_{ecto}$−3xFLAG lacking SPRING$_{ecto}$ (Supplementary Fig. 1j, k). A Michaelis-Menten analysis of the SPRING$_{ecto}$-S1P$_{ecto}$ protease activity showed the enzyme has an apparent $K_m$ of about 20 μM and a $K_{cat}$ of about 150 mol product per min per mol enzyme (Fig. 2d). This efficiency is considerably greater than a previously reported $K_m$ of ~100 μM and $K_{cat}$ of 0.16 mol product per min per mol enzyme measured for S1P$_{ecto}$ (lacking exogenous SPRING) using a peptide corresponding to S1P's B-site motif[21]. Therefore, we undertook structural studies to elucidate how SPRING enables S1P to cleave intermolecular substrates.

## Structural basis of the SPRING-S1P interaction

We determined the structure of the SPRING$_{ecto}$-S1P$_{ecto}$ complex to 2.3 Å using cryo-electron microscopy (cryo-EM) (Supplementary Fig. 2a−e and Table 1). The map accommodates the alphafold 2 (AF2) predictions[49] for the C-form of S1P with minor modifications and about half of the AF2 prediction for the SPRING ectodomain could be confidently placed (Fig. 3a and Supplementary Fig. 2f, g). This structural analysis revealed that the matured C-form of S1P$_{ecto}$ contains three subdomains that each interact with the other two subdomains to produce a heart-shaped holoenzyme (Fig. 3b, c). Beginning from the N-terminus, C-form S1P$_{ecto}$ contains a subdomain that belongs to the Peptidase S8 superfamily (residues 187−478)[3]. This domain contains a seven-strand parallel beta sheet with two large helixes above the sheet supporting the SPRING ectodomain (Fig. 3d and Supplementary Fig. 3a). The Peptidase S8 protease harbors a catalytic triad of D218, H249, and S414 and the oxanion hole residue N338[3,16]. D218 is positioned by the beta strand 1 (β1) whereas S414 and H249 are supported on two helices situated underneath the seven-stranded sheet. N338 is not located on an element of secondary structure but instead is on the

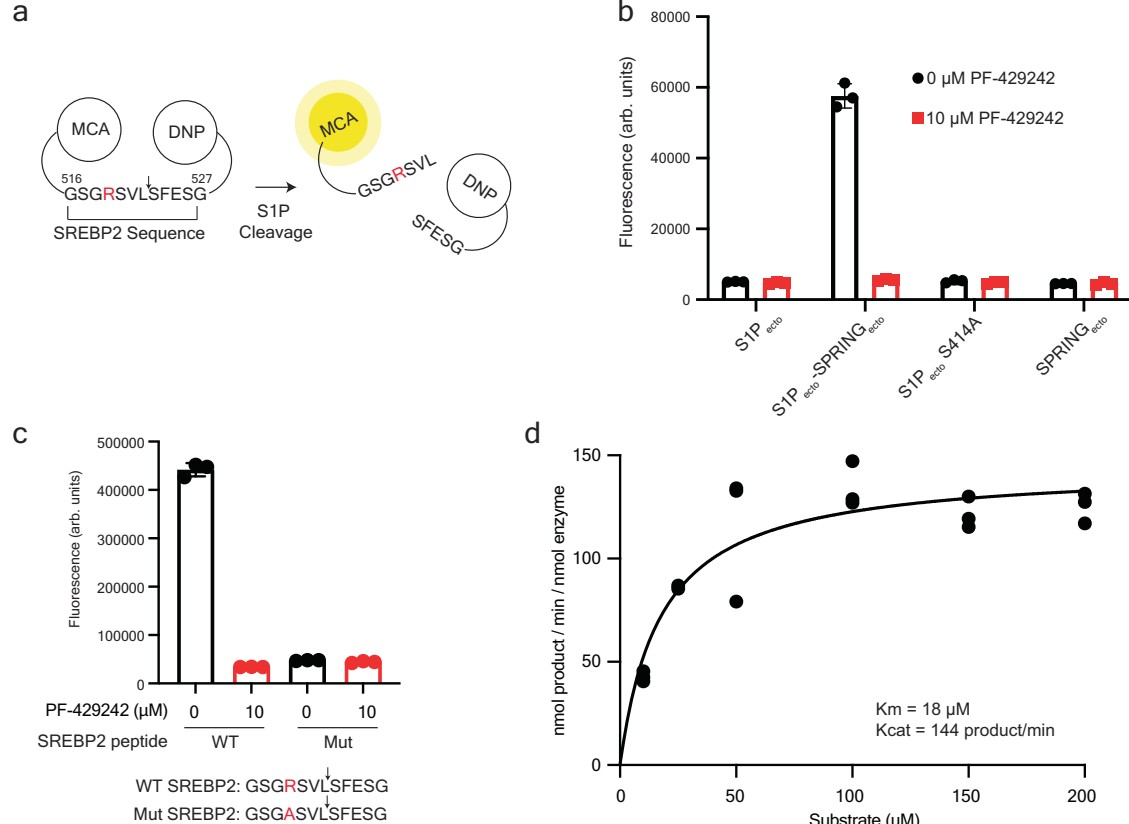

**Fig. 2 | SPRING$_{ecto}$ enhances the protease activity of S1P$_{ecto}$. a** Schematic for fluorescent S1P$_{ecto}$ protease assay based on the S1P cleavage motif in SREBP2. The P4 Arg is highlighted red. MCA = Methyl Cumaryl Amide DNP = 2, 4-dinitrophenyl. **b** Fluorescent SREBP2 cleavage assay. 100 nM of FLAG-S1P$_{ecto}$-His$_{10}$, the S1P$_{ecto}$-3xFLAG/SPRING$_{ecto}$-His$_{10}$ complex, catalytic mutant FLAG-S1P$_{ecto}$$^{S414A}$-His$_{10}$, or SPRING$_{ecto}$-His$_{10}$ was incubated with 50 μM SREBP2 peptide for 30 min at RT before measurement. Samples contained 10 μM PF-429242 or DMSO vehicle control as indicated. Assay was conducted with three technical replicates ($n = 3$). Error bars are standard error of the mean. Representative of at least three independent experiments. **c** Substrate specificity for the S1P$_{ecto}$-SPRING$_{ecto}$ complex. 50 nM of

S1P$_{ecto}$-3xFLAG/SPRING$_{ecto}$ was incubated with 50 μM of the indicated peptide for 30 min at RT before the fluorescence was read. Samples contained 10 μM PF-429242 or DMSO vehicle control as indicated. Error bars are standard error of the mean. Assay was measured in triplicate. Representative of at least three independent experiments. **d** S1P$_{ecto}$-SPRING$_{ecto}$ initial velocities graphed against substrate concentration. Each data point is the mean value of an independent experiment that was conducted with three technical replicates ($n = 3$). Three biological replicates are shown except for the 25 μM substrate concentration, which was included for two biological replicates. Curve fit calculated with Prism. Source data are provided as a Source Data file.

β5-β6 loop (Fig. 3d and Supplementary Fig. 3a). These four residues are well-resolved in our cryo-EM map (Fig. 3e). In many other members of this family of secretory pathway proteases, the protease is followed by an eight-strand beta sandwich subdomain called a homo B or P-domain that is necessary for folding the protease[20]. This subdomain was not previously identified in S1P, but the AF2 prediction and our structure reveal a beta sandwich subdomain in a similar position next to the protease subdomain that likely has a similar function (residues 479–616) (Fig. 3c). This domain assumes an overall fold like the P domain from Furin, but the topology is very different. The non-canonical fold is highlighted by a discontinuous N-terminal beta motif that participates in both sheets (Supplementary Fig. 3b, c). We refer to this subdomain as the "P-like subdomain." This domain is tightly wedged against the protease subdomain and protects a hydrophobic surface. The C-terminal portion of S1P$_{ecto}$ contains a poorly-characterized subdomain that has been identified in intraflagellar transport machinery, oligosaccharyltransferase pathways, and bacterial gliding proteins (GIFT subdomain, residues 617–998)[50] (Fig. 3b). The GIFT subdomain bears a passing resemblance to the protease subdomain and contains a major seven-strand beta sheet supported above and below by alpha helices. However, in contrast to the peptidase S8 subdomain, one of the beta strands are anti-parallel. (Supplementary Fig. 3d). The resolvable c-terminal residues of the GIFT subdomain (P925-A933) extend back over the top of the P-like

subdomain. The GIFT and Peptidase S8 subdomains interact through a mesh of loops that support, among other residues, the oxyanion hole residue N338 (see below). The interaction between the GIFT and Peptidase S8 subdomains also creates a negative electrostatic surface visible from the top of the enzyme (Fig. 3f) that extends to the front of the protein near the active site residues and may participate in controlling access to the enzyme by substrate proteins (Fig. 3g and see below).

Although enzymes in this family require calcium for activity[1], we did not observe obvious density for calcium within the protease subdomain. We did observe density for a putative calcium molecule in the GIFT subdomain within an acidic pocket near the GIFT-Protease interface (Supplementary Fig. 3e). The structural role of calcium is consistent with crystallographic studies characterizing the role of calcium in Furin, where the calcium supports the structure of the oxyanion hole residue but does not appear to directly act in catalysis[51].

Our cryo-EM map for S1P$_{ecto}$-SPRING$_{ecto}$ shows density for a 7-sugar tree at S1P N515, the form of N-linked glycan that is homogenously produced in HEK GnTI$^-$ cells. SPRING$_{ecto}$ appears to contact the terminal glycans of the tree, which could be best visualized using the unsharpened map (Supplementary Fig. 3f). We note that only the first three sugars with strong density are modeled in the final structure, but that the rest of the tree is placed here for illustration. This glycan plays an important structural role at the interface between the

## Table 1 | Cryo-EM data collection, refinement, and validation statistics

| | SPRING-S1P (EMDB-42639) (PDB 8UW8) | S1P (EMDB-42661) (PDB 8UWC) |
|---|---|---|
| **Data collection and processing** | | |
| Magnification | 81 kx | 165 kx |
| Voltage (kV) | 300 | 300 |
| Electron exposure (e–/Å$^2$) | 48 | 55 |
| Defocus range (μm) | –1.0 to –2.0 | –0.6 to –1.6 |
| Pixel size (Å) | 1.08 | 0.738 |
| Symmetry imposed | C1 | C1 |
| Initial particle images (no.) | 14,946,678 | 7,872,456 |
| Final particle images (no.) | 452,453 | 659,523 |
| Map resolution (Å) | 2.30 | 2.27 |
| FSC threshold | 0.143 | 0.143 |
| Map resolution range (Å) | 2.07–30.57 (mean 2.50) | 1.94–28.42 (mean 2.92) |
| **Refinement** | | |
| Initial model used (PDB code) | Alphafold (uniprot Q14703 and Q9H741) | Alphafold (uniprot Q14703) |
| Model resolution (Å) | 2.7 | 2.6 |
| FSC threshold | 0.5 | 0.5 |
| Model resolution range (Å) | n/a | n/a |
| Map sharpening B factor (Å$^2$) | –70 | –62 |
| Model composition | | |
| Non-hydrogen atoms | 6666 | 3849 |
| Protein residues | 835 | 494 |
| Ligands | 5 | 1 |
| B factors (Å$^2$) | | |
| Protein | 50.67 | 25.61 |
| Ligand | 50.78 | 30.37 |
| R.m.s. deviations | | |
| Bond lengths (Å) | 0.004 | 0.007 |
| Bond angles (°) | 0.595 | 0.906 |
| Validation | | |
| MolProbity score | 2.00 | 2.12 |
| Clashscore | 12.82 | 11.59 |
| Poor rotamers (%) | 0.14 | 1.67 |
| Ramachandran plot | | |
| Favored (%) | 94.44 | 94.63 |
| Allowed (%) | 5.56 | 5.37 |
| Disallowed (%) | 0.00 | 0.00 |

Peptidase S8 subdomain and the P-like subdomain. The first N-acetylglucosamine (GlcNAc) is mostly buried and forms a CH–π interaction[52] with S1P$_{ecto}$ Y561 (Supplementary Fig. 3g). Interestingly, this N-linked glycosylation is situated between the protease subdomain and P-like subdomain in a position similar to a druggable allosteric pocket that was identified on PCSK9[53] (Supplementary Fig. 3h, i). Mutation of N515 to Gln greatly reduced the autocatalytic maturation of secreted S1P$_{ecto}$, confirming that this residue (and by extension, the P-like subdomain) is critical for folding S1P$_{ecto}$ (Supplementary Fig. 3j).

SPRING$_{ecto}$ presents a cysteine-rich fold that we term the Cysteine-rich enabler of S1P activity (CREST) domain (residues 36-205). The AF2 prediction for its ectodomain returned no structural matches in the PDB using the DALI server[54] and searches in the FoldSeek database only identified SPRING proteins from other species. SPRING$_{ecto}$ is predicted to have seven intramolecular disulfide bonds,

and the concentration of these residues (15 cysteines within 95 modeled residues, including eight cysteines within a 20 amino acid stretch) precluded the assignment of unique pairs using mass spectrometry analysis of digested peptides. Given this complexity, the lack of experimental homology models, and the relatively lower local resolution for SPRING$_{ecto}$, we made only conservative adjustments of the initial AF2 model, which fit the density remarkably well (Supplementary Fig. 2f). The final model for SPRING$_{ecto}$ contains residues 76–138 and residues 165–196. The refined model and initial AF2 model are compared in Fig. 3h. Superposing the AF2 prediction with our modeled CREST domain in the context of the S1P ectodomain shows that the regions of the CREST domain that were not resolved in the map are likely flexible due to lack of contacts with S1P$_{ecto}$ (Fig. 3i), although we cannot exclude denaturation resulting from interactions with the air-water interface. Interestingly, the portion of SPRING interacting with S1P does not contain many secondary structural elements, suggesting the observed structure is stabilized by the extensive network of disulfide bonds (Fig. 3h). This may also contribute to heterogeneity in the map. Nevertheless, the residues interacting with S1P could be confidently modeled and analysis of the B-factors for the refined residues does not show any dramatic change between the C-domain of S1P and the CREST domain of SPRING (Supplementary Fig. 3k). SPRING$_{ecto}$ makes protein-protein contacts with all three subdomains of C-form S1P$_{ecto}$ and the total binding surface between S1P$_{ecto}$ and SPRING$_{ecto}$ is ~1200 Å$^2$. SPRING$_{ecto}$ binds a somewhat hydrophobic ridge beginning at the GIFT subdomain, running along the protease, and terminating at the P-like subdomain (Fig. 3f).

To test the accuracy of our model, we made conservative Ala mutations to several residues in the SPRING$_{ecto}$-S1P$_{ecto}$ interface and tested them in a battery of experiments based on the secreted ectodomains (Fig. 4a). We carried out initial studies using our simple co-expression assay where SPRING$_{ecto}$ is cleaved by S1P$_{ecto}$ (i.e. Fig. 1e). This format has two advantages. Firstly, by relying on secreted proteins, we could exclude mutations that grossly misfold either protein. Secondly, we could monitor the auto-catalysis of S1P$_{ecto}$ and identify mutants that fail to cleave SPRING$_{ecto}$ due to their own failure to mature. We identified three residues on each protein that reduced the cleavage of SPRING$_{ecto}$ by S1P$_{ecto}$. These residues are depicted in Fig. 4b. Based on initial experiments, we selected individual mutants or combinations of mutants to achieve the strongest loss-of-cleavage phenotypes. As previously shown, S1P$_{ecto}$ cleaves SPRING$_{ecto}$ when co-expressed (Fig. 4c lane 2). The inactive S1P$_{ecto}$$^{S414A}$ does not cleave SPRING$_{ecto}$ (Fig. 4c lane 3). Two residues, SPRING$_{ecto}$ R174 and S1P$_{ecto}$ N328, are in the center of the interaction surface and either SPRING$_{ecto}$$^{R174A}$ or S1P$_{ecto}$$^{N328A}$ individually reduced cleavage of SPRING$_{ecto}$ by S1P$_{ecto}$ (Fig. 4c, lanes 4 and 5). The other four residues that were evaluated are peripheral to SPRING$_{ecto}$ R174 and S1P$_{ecto}$ N328 and highlighted two potential points in the interaction that could be disrupted by mutagenesis (Fig. 4b). S1P$_{ecto}$ P317 is in the Peptidase S8 subdomain and caps the end of a helix that makes several contacts with SPRING$_{ecto}$. S1P$_{ecto}$ W556 is a surface tryptophan within the P-like subdomain that makes non-polar contacts with SPRING$_{ecto}$. SPRING$_{ecto}$ R95 interacts with S1P$_{ecto}$ near S1P$_{ecto}$ W556, and SPRING$_{ecto}$ V180 interacts with S1P$_{ecto}$ P317. To obtain a robust phenotype, we co-expressed these point mutants in pairs to concomitantly target two different points in the interaction surface. The combination of SPRING$_{ecto}$$^{R95A}$ and S1P$_{ecto}$$^{P317A}$ or SPRING$_{ecto}$$^{V180A}$ and S1P$_{ecto}$$^{W556A}$ greatly reduced the cleavage of SPRING$_{ecto}$ while leaving the autocatalytic processing of S1P$_{ecto}$ intact (Fig. 4c lanes 6 and 7).

Based on the cleavage results, we carried out a pulldown experiment to test whether these structure-guided mutations decreased the interaction between SPRING$_{ecto}$ and S1P$_{ecto}$. HEK 293 T cells were transfected with HA-tagged SPRING$_{ecto}$ alone (Fig. 4d lane 1), FLAG-S1P$_{ecto}$-His$_{10}$ alone (Fig. 4d lane 2), or else co-transfected with pairs of WT or mutant ectodomains, as indicated (Fig. 4d lanes 3-8). In contrast

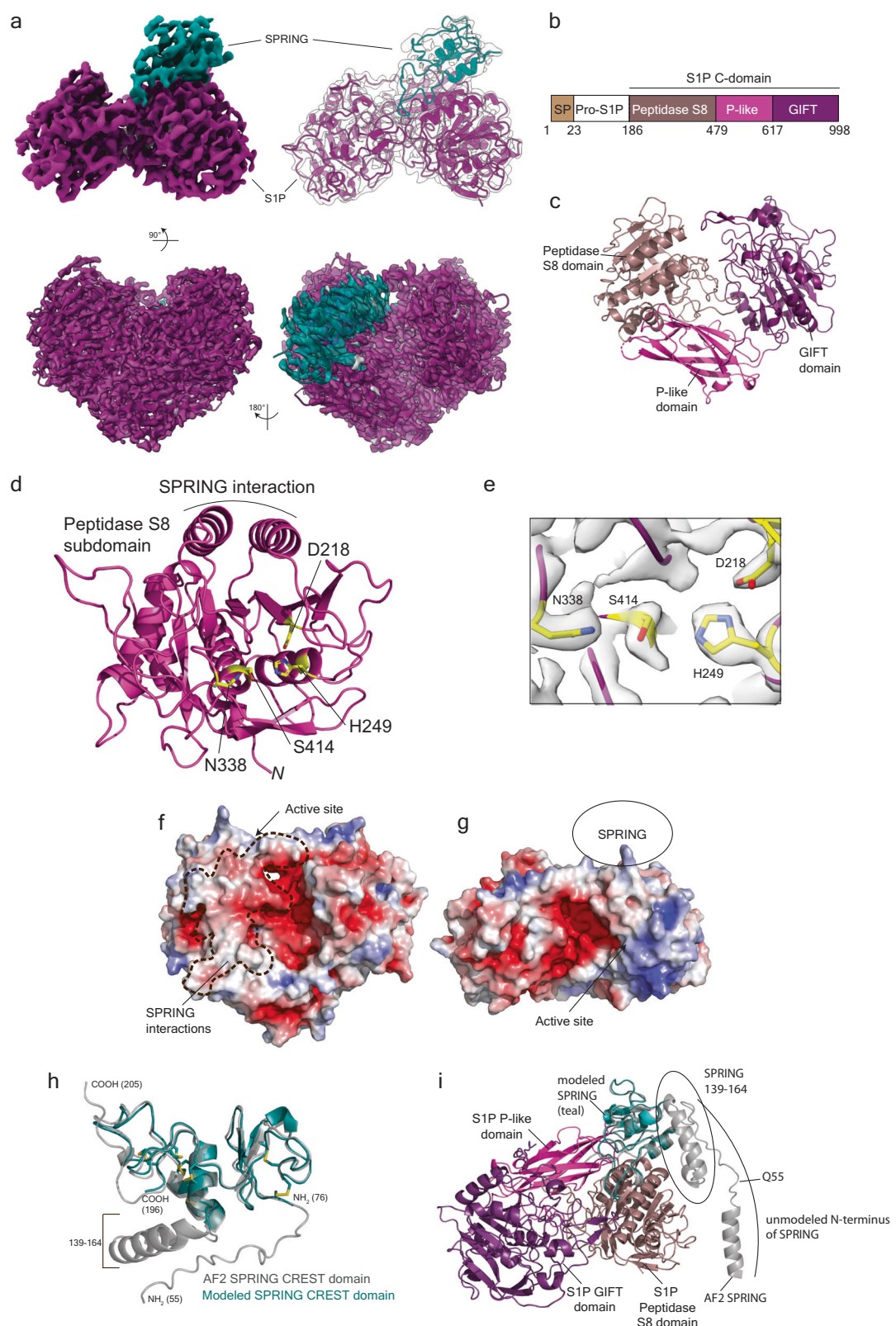

to WT S1P$_{ecto}$, S1P$_{ecto}$^S414A was unable to pulldown SPRING$_{ecto}$, showing this loss of interaction between the catalytically inert S1P$_{ecto}$ and SPRING$_{ecto}$ is consistent regardless of which interaction partner is being precipitated (Fig. 4d, lanes 3 and 4 and see Fig. 1i). Co-expression of S1P$_{ecto}$^P317A and SPRING$_{ecto}$^R95A or else S1P$_{ecto}$^W556A and SPRING$_{ecto}$^V180A abolished the interaction with SPRING$_{ecto}$-HA (Fig. 4d lanes 5,6).

Similarly, S1P$_{ecto}$^N328A was unable to precipitate SPRING$_{ecto}$ and SPRING$_{ecto}$^R174A was unable to be precipitated by S1P$_{ecto}$ (Fig. 4d, lanes 7 and 8). These results show the S1P$_{ecto}$-SPRING$_{ecto}$ interaction can be disrupted by structure-guided mutagenesis.

We asked whether our findings from the secreted ectodomains of S1P and SPRING faithfully capture the interaction between the full-length

**Fig. 3 | Cryo-EM structure of S1P$_{ecto}$-SPRING$_{ecto}$. a** Cryo-EM structure and model of S1P$_{ecto}$-SPRING$_{ecto}$ in different views. S1P map and model are colored purple and SPRING$_{ecto}$ is colored teal. Top panels depict the unsharpened map contoured at level 0.25 in ChimeraX. Bottom panels depict the sharpened map contoured at level 0.5 in ChimeraX. **b** Schematic for S1P domains highlighting the subdomains of the S1P C-domain. **c** S1P$_{ecto}$ C-domain colored by subdomains. View is the same as lower right panel in (**a**). Figure generated using Pymol. **d** Peptidase S8 domain of S1P shown in magenta cartoon. Catalytic residues depicted with yellow sticks. Helices involved in SPRING binding are labeled. N-terminus labeled N. **e** Density for S1P$_{ecto}$ active site residues. Map is the sharpened map contoured at level 0.8 in ChimeraX. Side chains for the catalytic residues are labeled and shown as yellow sticks. **f** S1P$_{ecto}$ C-domain solvent-accessible surface colored by electrostatic potential using the APBS plugin in Pymol. Scale is −5 kT/e (red) to +5 kT/e (blue). SPRING$_{ecto}$-binding surface is highlighted with dashed line. View is the same as lower right panel in (**a**). Figure generated using Pymol. **g** S1P$_{ecto}$ C-domain solvent-accessible surface colored as in (**f**) and viewed as if facing the active site. View is the same as the top left panel in (**a**). Position of SPRING is labeled with circle. Figure generated using Pymol. **h** AF2 model for the SPRING$_{ecto}$ ectodomain (grey) superposed with the refined model (teal). Termini of each model are indicated with NH$_2$ or COOH. A helical insert consisting of residues 139–164 was not resolved in the S1P-bound SPRING$_{ecto}$. Figure generated using Pymol. **i** AF2 model for SPRING (grey) superposed with modeled SPRING (teal) on the S1P structure (domains colored as in **c**). Figure generated using Pymol.

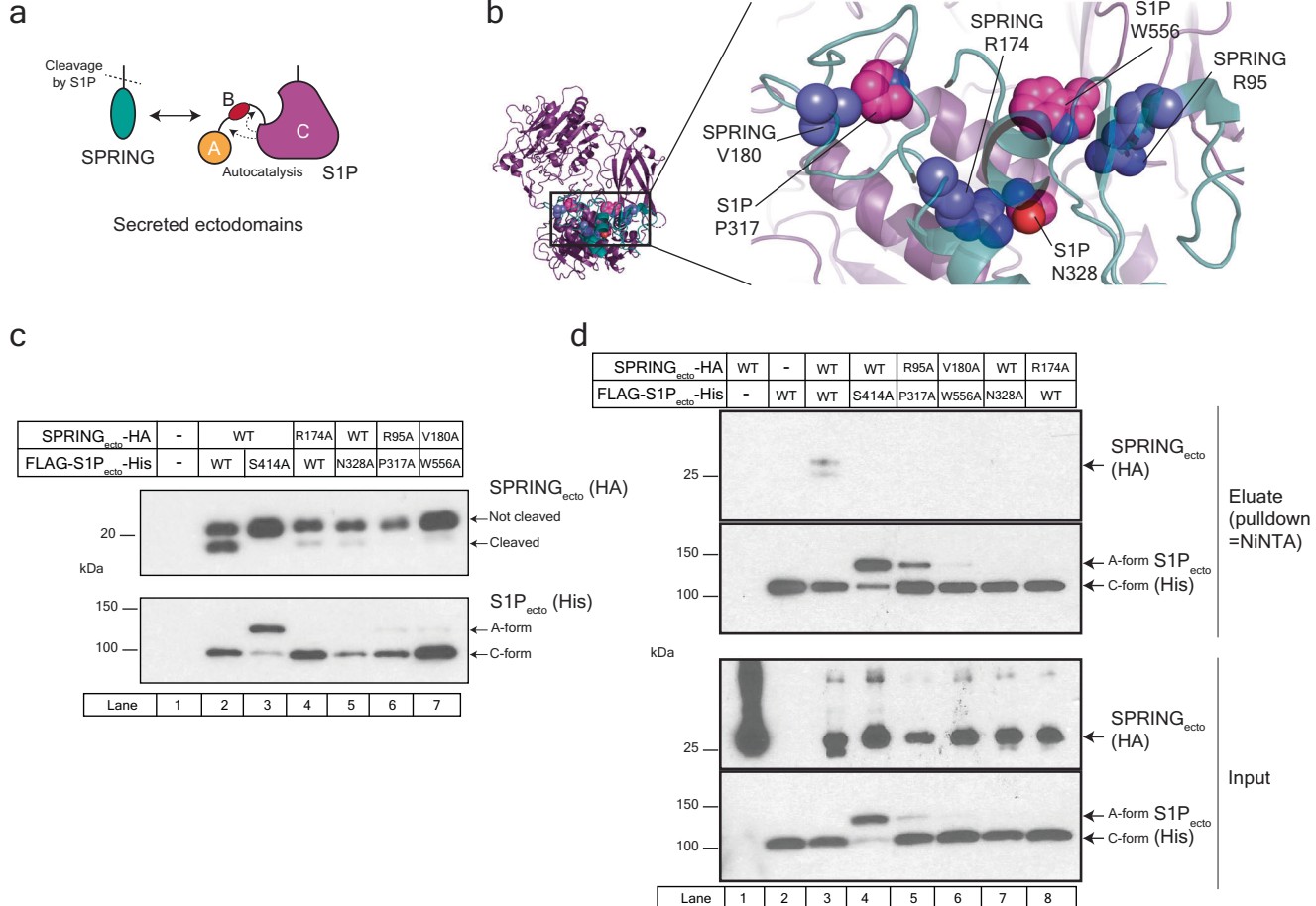

**Fig. 4 | Characterizing the interaction between the secreted ectodomains of S1P and SPRING. a** Cartoon depicting the interaction between the secreted ectodomains of SPRING and S1P. **b** Key molecular contacts between SPRING$_{ecto}$ and S1P$_{ecto}$. Side chains of residues investigated through mutagenesis are shown as spheres. SPRING is teal cartoon and blue spheres, S1P is magenta cartoon and magenta spheres. **c** SPRING$_{ecto}$ cleavage assay. HEK 293 T cells were transfected empty pEZT vector (lane 1) or else with the indicated plasmids. After 72 h, supernatants were collected and analyzed by immunoblot. SPRING$_{ecto}$ was detected using anti-HA antibodies and S1P$_{ecto}$ was detected using anti-His antibodies. **d** Pulldown of SPRING$_{ecto}$ by S1P$_{ecto}$. HEK 293 T cells were transfected with empty pEZT vector, or the constructs as indicated. After 72 h, supernatants were subjected to NiNTA pulldown and analyzed by immunoblot. SPRING$_{ecto}$ was detected using anti-HA antibodies and S1P$_{ecto}$ was detected using anti-His antibodies. All immunoblots are representative of at least 3 independent experiments. Source data are provided as a Source Data file.

proteins (Fig. 5a). First, we expressed full-length S1P (S1P$_{FL}$) harboring a C-terminal V5 epitope tag with or without full-length SPRING (SPRING$_{FL}$) harboring a C-terminal Myc tag. S1P$_{FL}$-V5 expressed without SPRING$_{FL}$-Myc produces three bands corresponding to the A, B, and C-forms of S1P (Fig. 5b, lane 2). The A-form is dominant, and the C-form appears weakest. Co-expression of S1P$_{FL}$-V5 with SPRING$_{FL}$-Myc creates a different S1P expression pattern. The B-form is no longer detected, and the two major bands are the A-form and the C-form (Fig. 5b, lane 4). This altered expression pattern is consistent with previous reports[36,39].

Next, we tested whether the glycosylation mutant N515Q affected S1P$_{FL}$. This assay introduces another important readout: the cleavage of SPRING$_{FL}$ by S1P$_{FL}$ results in the secretion of SPRING into the culture media[39] (Fig. 5c lane 1). As presaged by our experiments with S1P$_{ecto}$, S1P$_{FL}^{N515Q}$ resembles the catalytically inert S1P$_{FL}^{H249A}$ in that it cannot mature nor cleave SPRING$_{FL}$ (Fig. 5c lanes 2-3). Finally, we tested whether the structure-guided mutations to SPRING$_{FL}$ or S1P$_{FL}$ impaired the functional interaction between SPRING$_{FL}$ and S1P$_{FL}$. Indeed, introduction of the structure-guided mutations into S1P$_{FL}$ or SPRING$_{FL}$ reduced the

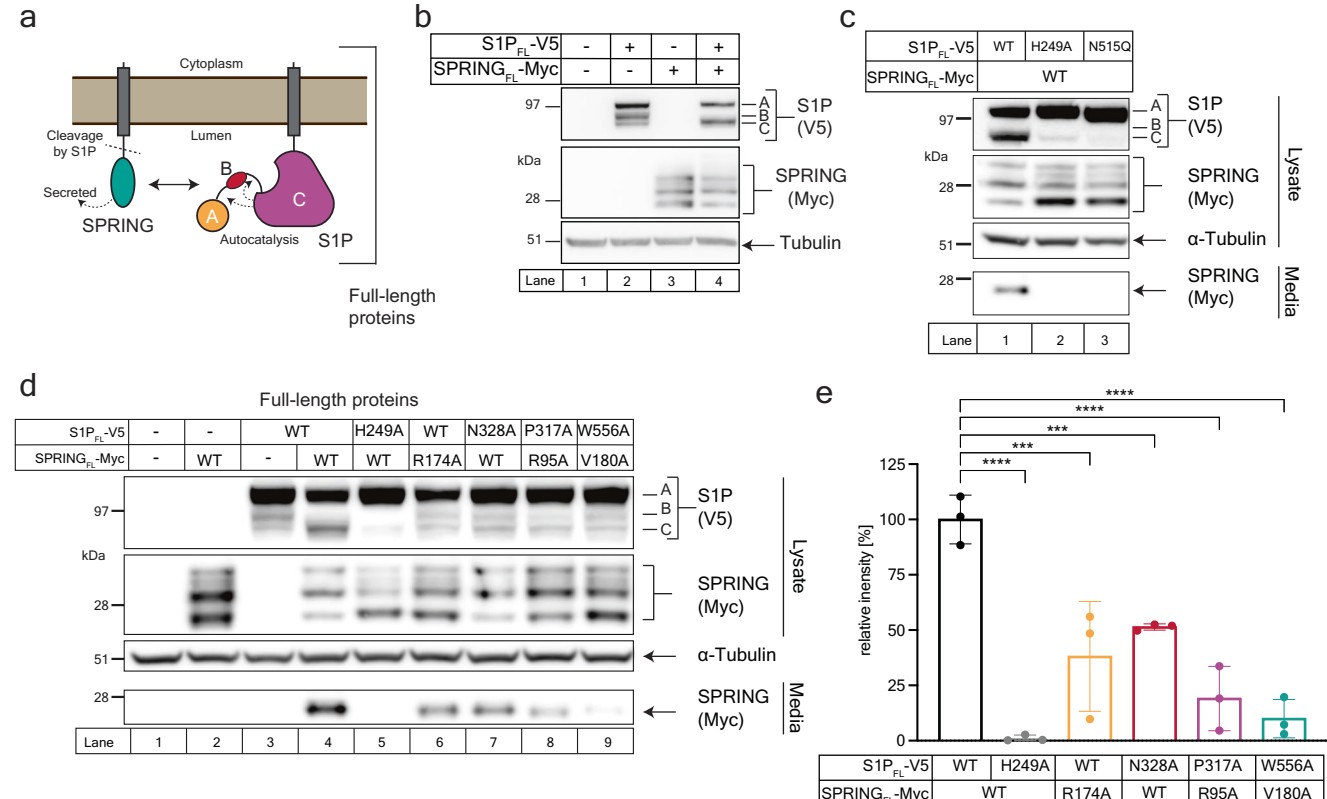

**Fig. 5 | Characterizing the interaction between full-length S1P and SPRING.**
**a** Schematic for the interaction between full-length S1P and SPRING. **b** SPRING regulates S1P maturation. HEK 293 T cells were transfected with the indicated plasmids and the lysates analyzed by immunoblot as described in *Methods*. S1P was detected using anti-V5 antibodies, SPRING was detected using anti-Myc antibodies, and alpha-Tubulin was detected using anti-Tubulin antibodies. **c** Critical role of S1P N515 on S1P maturation. HEK 293 T cells were transfected with the indicated plasmids. Lysates and media samples were analyzed by immunoblot. S1P was detected using anti-V5 antibodies, SPRING was detected using anti-Myc antibodies, and alpha-Tubulin was detected using anti-Tubulin antibodies. **d** Effect of SPRING and S1P mutants on S1P maturation and processing of SPRING. HEK 293 T cells were

transfected with the indicated plasmids. Lysates and media samples were analyzed by immunoblot. S1P was detected using anti-V5 antibodies, SPRING was detected using anti-Myc antibodies, and alpha-Tubulin was detected using anti-Tubulin antibodies. **e** Quantification of SPRING detected in the media in three independent experiments carried out as in (**d**). Statistical significance was tested using one-way ANOVA with Holm-Sidak post hoc analysis. Bars and errors represent mean ± SD; ***$p < 0.001$, ****$p < 0.0001$. The adjusted *P*-values for SPRING/S1P vs mutants are 0.000004 for SPRING/S1P$^{H249A}$, 0.000173 for SPRING$^{R174A}$/S1P, 0.000674 for SPRING/S1P$^{N328A}$, 0.00002 for SPRING$^{R95A}$/S1P$^{P317A}$, and 0.000009 for SPRING$^{V180A}$/S1P$^{W556A H249A}$. All immunoblots are representative of at least 3 independent experiments. Source data are provided as a Source Data file.

maturation of S1P$_{FL}$ such that the S1P$_{FL}$ expression pattern mimics that of S1P$_{FL}$ expressed without SPRING$_{FL}$ (Fig. 5d compare lane 3-4 to lanes 6-9). Additionally, all the mutants resulted in decreased secretion of SPRING (Fig. 5d). Densitometry analysis of secreted SPRING from three independent experiments confirmed that the structure-guided mutations impair the secretion of SPRING (Fig. 5e). These results strongly suggest the interaction we identified between the ectodomains of S1P and SPRING is critical for the SPRING-mediated activation of S1P.

**SPRING stabilizes inter-domain interactions in S1P**
We interpret the structure of S1P$_{ecto}$ bound to SPRING$_{ecto}$ as representing the matured form of that complex. To better understand how SPRING activates S1P, we wanted to understand the fate of S1P$_{ecto}$ when it is matured in the absence of SPRING$_{ecto}$. We therefore obtained a cryo-EM structure of the S1P$_{ecto}$ purified without co-expressed SPRING$_{ecto}$ (Supplementary Fig. 4 and Table 1). The map had a resolution of 2.27 Å using GSFSC cutoff of 0.143 and shows the A-domain bound atop the Peptidase S8 subdomain in a similar position as SPRING$_{ecto}$ (Fig. 6a). The A pro-domain is a compact unit consisting of an anti-parallel four-stranded beta sheet supporting two alpha helices within the β1-β2 and β3-β4 loops (Supplementary Fig. 5a). The N- and C-termini of the A-domain run along the side of the Peptidase S8 subdomain. This structure is from a WT S1P$_{ecto}$ sample that has undergone autocatalysis (Fig. 1g) and therefore the density sharply disappears after K137, which

corresponds to cleavage at the S1P B-site motif (Fig. 6b). Despite the nominally high global resolution, the GIFT subdomain was not visible in the reconstruction, which we interpret to indicate flexibility of this domain relative to the other subdomains.

Our structure of S1P$_{ecto}$ shows that the B-site motif at the C-terminal end of the A-domain is occupying the active site. The P8 residue R130 and P4 residue R134 both make favorable contacts in acidic pockets on the Peptidase S8 subdomain (Fig. 6c). (R130 can be utilized as the P4 residue for the alternative B' motif[23]). Particularly strong density for the P1-6 residues is visible in the map (Fig. 6d). The P6 Val also makes hydrophobic contacts and the P4 Arg is inserted deeply into an acidic pocket where it makes a salt bridge with D218 of the catalytic triad as well as Q282 and S307 (Fig. 6e). These polar contacts provide an explanation for the requirement of the basic Arg at this position. The fact that the B-site motif uses the P6 and P8 residues to make additional contacts outside of the canonical four-residue motif likely contributes to the difficulty the SREBP2 peptide faces when competing for this site as the SREBP2 sequence (S$_8$GSGR$_4$SVL ↓) lacks the favorable P6 and P8 residues of the S1P B-site motif (R$_8$KVFR$_4$SLK ↓).

Unlike SPRING$_{ecto}$, the A-domain of S1P$_{ecto}$ appears to interact only with the Peptidase S8 subdomain (Fig. 6a). It does not contact the P-like subdomain, and any contacts with the GIFT subdomain, if they take place, are not stable enough to be resolved. Despite an overall RMSD of 0.94 Å between the resolvable residues of S1P from the

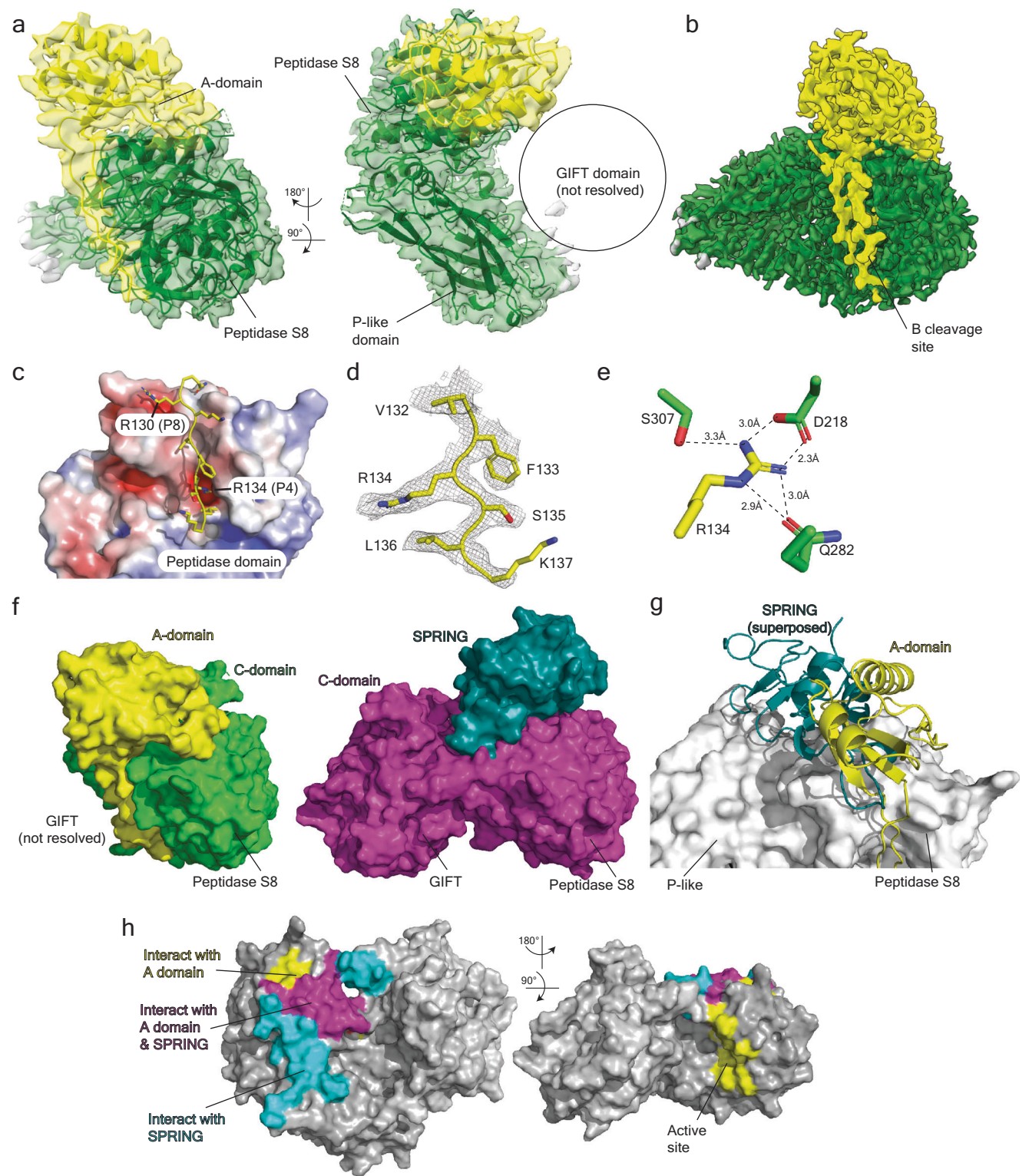

structures of S1P$_{ecto}$ and S1P$_{ecto}$-SPRING$_{ecto}$, the globular portion of the S1P A-domain would severely clash with SPRING (Fig. 6f, g and Supplementary Fig 5b). This competition is remarkable given that SPRING$_{ecto}$ and the S1PA domain share little conserved sequence. There are no conserved structural elements between the A-domain of S1P and the CREST domain of SPRING and residues that can be aligned by sequence show obviously different structural roles (Supplementary Fig 5b, c). Nonetheless, the A-domain binds the Peptidase S8 sub-domain using an epitope that largely overlaps with the epitope used by SPRING$_{ecto}$ (Fig. 6h, left). However, unlike the A-domain, SPRING$_{ecto}$

does not bind the active site of S1P$_{ecto}$ (Fig. 6h, right). The overlapping epitopes for SPRING$_{ecto}$ and the S1P A-domain are further supported by comparing our cryo-EM structure of S1P$_{ecto}$-SPRING$_{ecto}$ to the AF2 prediction for full-length S1P without SPRING, which essentially matches our cryo-EM structure of S1P$_{ecto}$, but which also models the GIFT subdomain (Supplementary Fig. 5d–g). The flexibility of the GIFT subdomain in the absence of SPRING$_{ecto}$ appears connected to the stability of the oxyanion hole residue N338. Two loops of the Peptidase S8 subdomain (residues 338-348 and residues 372-384) pack against the GIFT subdomain (Fig. 7a) and were not resolved in the map of

**Fig. 6 | Cryo-EM structure of S1P$_{ecto}$ without SPRING$_{ecto}$. a** Cryo-EM map and model of S1P$_{ecto}$ in different views. S1P$_{ecto}$ map and model are colored green for the C-domain and yellow for the A-domain. **b** Cryo-EM map and model of S1P$_{ecto}$ highlighting the position of the B site cleavage motif on the S1P$_{ecto}$ Peptidase S8 subdomain. S1P$_{ecto}$ map and model are colored green for the C-domain and yellow for the A-domain. **c** Interaction of the B' and B-site residues with the Peptidase S8 subdomain. Peptidase S8 is shown as solvent-accessible surface colored by electrostatic potential using the APBS plugin in Pymol. Scale is -5 kT/e (red) to +5 kT/e (blue). B-site cleavage motif is yellow with side chains shown as sticks. **d** Density for S1P$_{ecto}$ residues 132-137. Sharpened map was contoured at level 10 and carved at 1.5 Å around the shown residues. **e** Details of the P4 pocket. R134 is shown as yellow sticks. Residues making electrostatic interactions with the R134 side chain are shown in green sticks and interaction distances are labeled. **f** SPRING and the A-domain bind the same epitope on S1P-C. The structures of S1P$_{ecto}$ and S1P$_{ecto}$-

SPRING$_{ecto}$ are shown in equivalent positions and depicted as molecular surfaces. SPRING$_{ecto}$ is colored teal and SPRING-bound S1P$_{ecto}$ is colored magenta. The A-domain from S1P$_{ecto}$ is colored yellow, and the C-domain from S1P$_{ecto}$ without SPRING$_{ecto}$ is colored green. **g** SPRING and the A-domain bind the same epitope on S1P-C. The Peptidase S8 and P-like subdomains of S1P$_{ecto}$ are depicted in grey surface. SPRING and the A-domain are shown as cartoons. SPRING$_{ecto}$ is colored teal and the A-domain from S1P$_{ecto}$ is colored yellow. SPRING was placed by superposing the Peptidase S8 domains of the two forms of S1P. **h** Overlapping interaction surfaces for SPRING$_{ecto}$ and the S1P$_{ecto}$ A-domain. S1P$_{ecto}$ from the S1P$_{ecto}$-SPRING$_{ecto}$ structure is shown as molecular surface. Residues interacting with the A-domain but not SPRING$_{ecto}$ are colored yellow. Residues interacting with SPRING$_{ecto}$ but not the A-domain are colored cyan. Residues interacting with both are colored magenta. Panels (**a, b**) were generated using ChimeraX, the other panels were generated using Pymol.

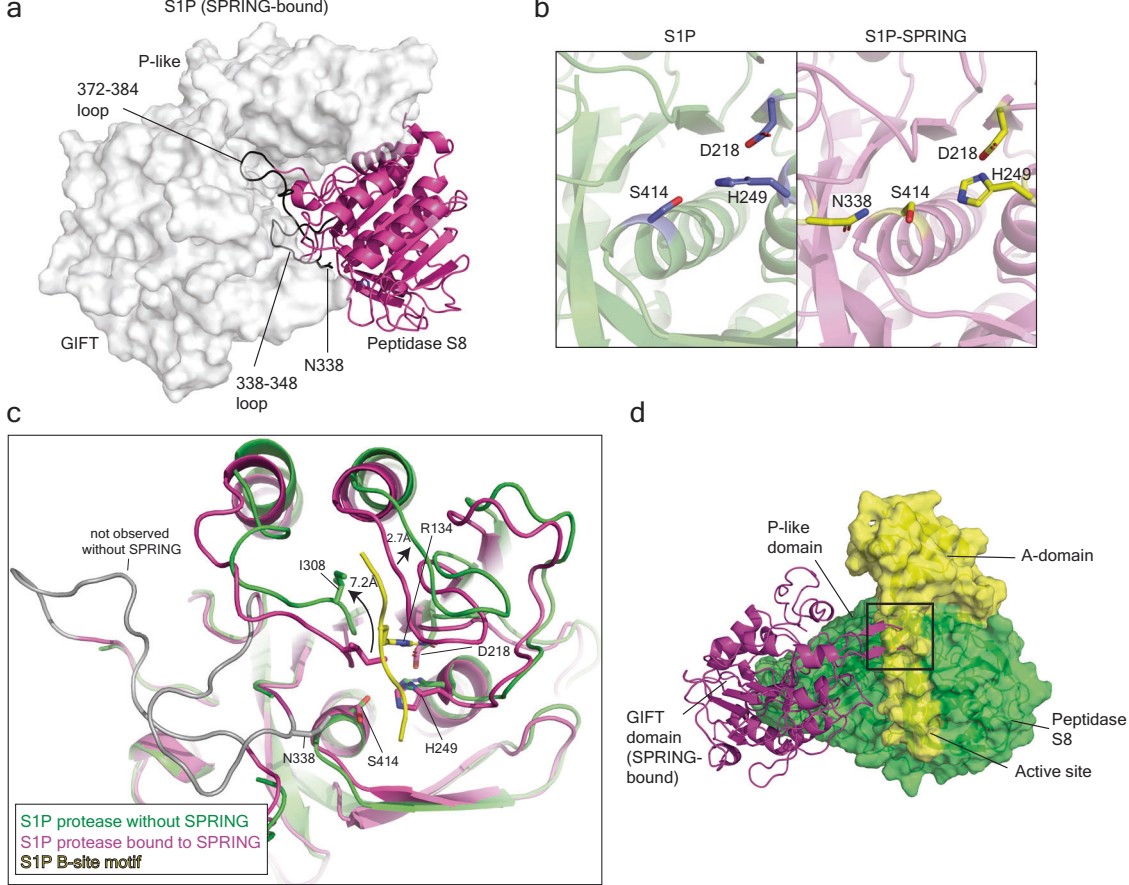

**Fig. 7 | Features of S1P associated with substrate recognition. a** The GIFT subdomain stabilizes loops of the Peptidase S8 subdomain. Structure of S1P in the SPRING-bound form shown where the Peptidase S8 domain is shown as magenta cartoon. The P-like and GIFT subdomains are shown as grey surface. Residues of the S1P Peptidase S8 subdomain that could not be modeled in the structure of S1P without SPRING are shown as grey cartoon. N338 is depicted as grey sticks. Figure generated using Pymol. **b** Comparison of the active sites from S1P$_{ecto}$ and S1P$_{ecto}$-SPRING$_{ecto}$. S1P$_{ecto}$ is depicted in green cartoon and modeled catalytic residues are shown as blue sticks. S1P$_{ecto}$ from the model of S1P$_{ecto}$-SPRING$_{ecto}$ is shown as purple cartoon and modeled catalytic residues are shown as yellow sticks. Figure generated using Pymol. **c** Conformation changes associated with substrate binding. The Peptidase S8 subdomains of S1P$_{ecto}$ bound to SPRING$_{ecto}$ and S1P$_{ecto}$ alone are

shown as cartoons colored magenta or green, respectively. Residues of the S1P$_{ecto}$ Peptidase S8 subdomain that could not be modeled in the structure of S1P$_{ecto}$ without SPRING are shown as grey cartoon. N338 is depicted as grey sticks. Catalytic residues and I308 are shown as sticks. S1P$_{ecto}$ residues 132–137 are shown in yellow and R134 is shown as sticks. Movement of loops are highlighted with black arrows. Figure generated using Pymol. **d** The GIFT subdomain in the SPRING-bound S1P$_{ecto}$ clashes with the A-domain in the S1P$_{ecto}$ without SPRING. S1P protease subdomains were superposed. The GIFT domain from S1P$_{ecto}$-SPRING$_{ecto}$ is depicted as magenta cartoon. The Peptidase and P-like subdomains from S1P$_{ecto}$ without SPRING$_{ecto}$ are shown as green surface, and the A-domain is shown as yellow surface. For clarity, SPRING and the Peptidase and P-like subdomains from S1P$_{ecto}$-SPRING$_{ecto}$ are omitted. Figure generated using Pymol.

S1P$_{ecto}$ without SPRING$_{ecto}$. In contrast, N338 is well-resolved in the map for S1P$_{ecto}$-SPRING$_{ecto}$ (Fig. 7b and Fig. 3e). The stabilization of N338 by the GIFT subdomain likely contributes to the increased protease activity of the S1P$_{ecto}$-SPRING$_{ecto}$ complex.

We compared the structure of S1P$_{ecto}$-SPRING$_{ecto}$ to that of S1P$_{ecto}$ for clues about the mechanism of substrate recognition by S1P. The structure of S1P$_{ecto}$ is bound to the product resulting from the cleaved B-site whereas the structure of S1P$_{ecto}$-SPRING$_{ecto}$ represents an apo

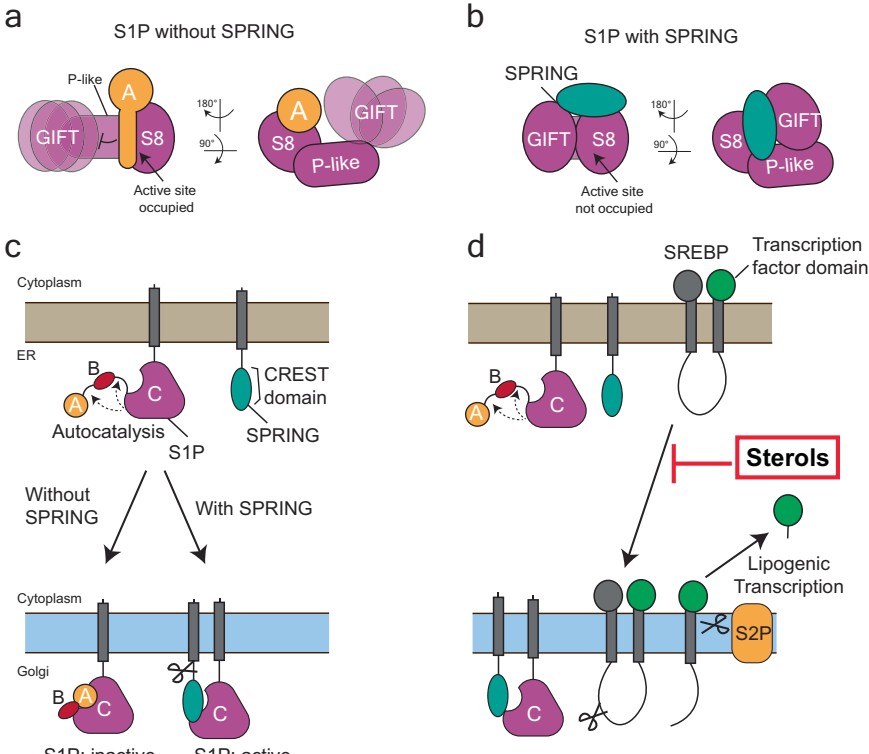

**Fig. 8 | Model for role of SPRING in S1P activation. a** Schematic for the state of S1P$_{ecto}$ matured without SPRING$_{ecto}$. **b** Schematic for the state of S1P$_{ecto}$ that is bound SPRING$_{ecto}$. **c** Schematic for maturation of S1P and its trafficking from the ER to the Golgi without or with SPRING. **d** Schematic for the role of SPRING in activating S1P to cleave SREBP. A = A-domain; B = B-domain, S1P site one protease, S2P site two protease, SREBP sterol response element binding protein; SPRING SREBP regulating gene.

form of the enzyme. Therefore, we restrict our analysis to changes associated with the presence of the B-site peptide (e.g. S1P residues 132-137, corresponding to the P6-P1 sequence). In Fig. 7c, the Peptidase domains of the two forms of S1P$_{ecto}$ are superposed. Residues 338-348 and 372-384 of S1P$_{ecto}$-SPRING$_{ecto}$ are colored grey because they are not modeled in the structure of S1P$_{ecto}$. In the modeled residues, two notable changes are the movement of I308 from a position in or near the P4 pocket in the apo structure (S1P$_{ecto}$-SPRING$_{ecto}$) to a new position away from the pocket in the product-bound S1P$_{ecto}$. The β-Carbon of I308 moves ~7.2 Å with the rearrangement of the 306-314 loop. The second change is an upward shift of ~3 Å of the loop consisting of residues 276–286. Unlike the I308 loop, this rearrangement is essentially a rigid movement. The rearrangement of these loops creates a groove for the B-site peptide (Fig. 7c). Finally, we note that loop from the GIFT subdomain that interacts with SPRING in the SPRING-bound form of S1P$_{ecto}$ clashes with residues 49-51 and 130-132 of the A-domain (Fig. 7d). This loop of the GIFT domain may control the accessibility of the active site as it would have to be repositioned for S1P$_{ecto}$-SPRING$_{ecto}$ to cleave substrates with a motif resembling the B-site of S1P.

## Discussion

The current study provides insights into how S1P matures and is licensed by SPRING to cleave external substrates. Without SPRING$_{ecto}$, S1P$_{ecto}$ undergoes autocatalysis but remains non-covalently associated with its inhibitory A-domain, as reported previously[20,22]. Consequently, this S1P$_{ecto}$ is essentially inactive in protease assays based on the S1P cleavage motif from SREBP2 (Fig. 2). In contrast, S1P$_{ecto}$ co-expressed with SPRING$_{ecto}$ forms a stable heterodimeric complex lacking the cleaved S1P A-domain (Fig. 1h) and efficiently cleaves an exogenous SREBP2 peptide (Fig. 2). Our biochemical and structural data suggest two mechanisms by which SPRING$_{ecto}$ activates S1P$_{ecto}$. First, SPRING$_{ecto}$ displaces the inhibitory S1P A-domain, freeing the S1P

active site from the B-site peptide and thereby granting access to cognate substrates. Second, SPRING$_{ecto}$ promotes inter-domain inter-actions in S1P that stabilize the oxyanion hole residue N338 at the active site (Fig. 8a, b). Based on our results here and other recent reports[36,39], we hypothesize that S1P undergoes A-form to B-form autocatalysis without SPRING. However, S1P requires SPRING to dis-place the A domain in order to cleave external substrates like SREBP2 (Fig. 8c, d). This is also reflected in increased B-form to C-form maturation (Fig. 5), since efficient cleavage of the C-site motif would require displacing the B-site peptide. Owing to the current study's reliance on over-expression, we did not explore the role of SPRING in S1P trafficking. Instead, we suggest that the critical role of SPRING is to license the proteolytic activity of S1P against SREBPs.

A comparison of the structures of S1P$_{ecto}$-SPRING$_{ecto}$ and S1P$_{ecto}$ to other subtilisin-like secretory serine proteases of the proprotein convertase family reveals that SPRING is likely to be specific for S1P (Supplementary Fig. 6). Using Furin (PDB: 4Z2A)[55] and PCSK9 (PDB 2PMW)[44] as examples, we observed that the beta-sandwich P-domain of Furin or the Cys-Rich Domain (CRD) of PCSK9 are both situated at a different angle to the protease subdomain than the P-like domain of S1P$_{ecto}$, which makes critical contacts with SPRING. The bound pro-domain of PCSK9 occupies the same position as the pro-domain of S1P$_{ecto}$ and would inhibit PCSK9 binding to the ectodomain of SPRING. Finally, the GIFT subdomain also makes contacts with SPRING$_{ecto}$, and this subdomain is distinct to S1P within the PCSK family[50]. Accordingly, SPRING does not promote the proteolytic maturation of either PCKS9 or Furin[39]. PCSK9 is an interesting comparison for S1P and S1P-SPRING. Like S1P, PCSK9 undergoes autocatalysis. However, PCSK9 does not release its cleaved pro-domain, which remains tightly integrated with the enzyme, rendering the PCSK9 enzyme functionally inactive against external substrates[43–45]. We propose that SPRING rescues S1P from the fate of PCSK9.

The experiment where $S1P_{FL}$ was expressed without and with $SPRING_{FL}$ (Fig. 5b) mirrors the expression pattern observed in over-expressed vs endogenous S1P from an earlier study (see Fig. 5a from ref. 40). In that report, expressing S1P in S1P-deficient cells resulted in a pattern like Fig. 5b lane 2, where the A, B, and C-forms of S1P were present. However, immunoblot analysis of endogenous S1P from WT cells showed a pattern like Fig. 5b lane 4, with endogenous S1P only detected in the A and C forms. Taken together, we propose that SPRING is a limiting co-factor for the autocatalytic maturation of S1P. In the absence of SPRING, S1P processing to the C′/C-sites will be impaired due to i) the presence of the B-site peptide in the active site and ii) the flexible oxyanion hole residue N338. This SPRING-dependent S1P expression pattern was also observed in another study where it was proposed that SPRING causes S1P to skip the B′/B sites and cleave itself directly at the C′/C-sites[36]. However, mutating the B′/B-sites cleavage site abolishes maturation of S1P[20], including in the presence of over-expressed SPRING[39]. Moreover, S1P can be cleaved at its C′/C-sites in trans[21] and appears prone to non-specific degradation in that region (Fig. 1g and refs. 22,47). Our interpretation is that SPRING accelerates the maturation of B-form S1P to C-form S1P and does not induce an alternate sequence of cleavage steps.

Our SREBP2 cleavage assay (Fig. 2) showed robust SREBP2 cleavage activity with the $S1P_{ecto}$-$SPRING_{ecto}$ complex but not with $S1P_{ecto}$ purified without $SPRING_{ecto}$. Similar protease assays have previously been used to detect activity with versions of $S1P_{ecto}$[21,42,47]. It is possible that SPRING is required for S1P to cleave some substrates but not others, or to modify its substrate scope. Indeed, the first characterization of purified S1P activity showed slow but measurable cleavage of a fluorescent peptide containing a 6-residue derivative of S1P B-site motif but no activity for a corresponding 6-mer SREBP2 peptide. Detecting SREBP2 cleavage required HPLC analysis following a 4 h, 37 °C incubation of the enzyme with a 16-mer SREBP2 peptide[21]. Another possibility is that S1P retains weak activity that was not detected in our experiments. Previous experiments, including the work that led to the discovery of PF-429242, typically incubated the enzyme for 4 h or longer at 37 °C[21,42,56], whereas our assay shows robust cleavage in minutes at RT by $S1P_{ecto}$-$SPRING_{ecto}$. We interpret this to mean that physiological cleavage of SREBP2 by S1P requires the activation of S1P by SPRING. Since SPRING contains an S1P cleavage site and can be secreted when cleaved by S1P (e.g. Figure 5), it is possible that previous experiments co-purified small amounts of endogenous SPRING ectodomains with the secreted, exogenous S1P. More experiments are needed to determine the substrate scope of S1P-SPRING vs S1P, and additional biological factors likely remain unknown. Although this study focused on peptides derived from SREBP2, we expect similar results would be observed with peptides derived from the conserved S1P cleavage motif in SREBP1 (RNVL in SREBP1 vs RSVL in SREBP2). Additionally, it will be interesting to test whether SPRING is required for the S1P-mediated maturation of viral glycoproteins from Arenaviruses and other S1P substrates[1,11,57,58]. Such studies may provide insights as to the cellular compartment in which SPRING and S1P interact. Finally, while our experiments took advantage of the ability of HEK 293 cells to produce soluble, secreted $S1P_{ecto}^{S414A}$ for biochemical studies, it is well-established that immature full-length S1P is retained in the ER[23].

Several questions remain to be answered. We do not know whether SPRING and S1P interact first in the ER or Golgi, although we and others have previously shown that a SPRING modified to include a KDEL ER retrieval sequence was unable to rescue SREBP signaling in SPRING-KO cells[35,37]. An attractive hypothesis is that SPRING interacts with S1P in the Golgi as part of a mechanism to ensure that S1P cleaves substrates only in the Golgi but not the ER. Additionally, the physiological function of the S1P cleavage motif on SPRING remains mysterious. Our study took advantage of this motif to assess the functional interaction between SPRING and S1P. Full-length S1P contains a C-terminal sheddase sequence[23]. It is possible that cleavage of SPRING

and the shedding of S1P provide a mechanism to terminate S1P activity. Finally, in humans, a Q55R variant in SPRING is associated with circulating levels of HDL cholesterol and ApoA1[41]. Unfortunately, this residue was not resolved in our structure. Our current study reveals SPRING dislodges the inhibitory S1P A-domain from the matured S1P and activates S1P by releasing this inhibition and licensing the enzyme for catalysis.

## Methods

### Chemicals and antibodies

PF-429242 dihydrochloride was purchased from Sigma-Aldrich (Cat # SML0667). Ni-NTA Superflow resin was purchased from Qiagen (Cat # 30430). X-tremeGENE HP was purchased from Sigma (Cat #26181). Anti-HA Agarose was purchased from Thermofisher (Cat #26181). Quantifoil R 1.2/1.3 300 Mesh, Au grids were purchased from Electron Microscopy Sciences. PNGaseF, EndoH, and HiFi DNA Assembly Mix were purchased from NEB.

Fluorescent peptides were synthesized by Genscript. The SREBP2 peptide MCA-GSGRSVLSFESG-Lys(DNP) was synthesized at ≥ 85% purity for the experiments depicted in Fig. 2b. This peptide and the mutant peptide MCA-GSGASVLSFESG-Lys(DNP) were re-synthesized at ≥ 91% for the experiment depicted in Fig. 2c, d and Supplementary Fig. 1j, k.

The following antibodies and dilutions were used in this study: anti-FLAG M2 clone (Sigma-Aldrich Cat # F1804, diluted 1:1000), HIS.H8 anti-His antibody (Sigma-Aldrich Cat # 05-949, diluted 1:1000), HA Tag Monoclonal Antibody (2-2.2.14) (ThermoScientific Cat # 26183, diluted 1:1000), V5-tag (Invitrogen Cat# R960-25, diluted 1:1000), anti-alpha-Tubulin (Sigma Cat# T9026, diluted 1:2000), HRP-conjugated anti-Myc tag antibody (Thermo Scientific Cat# R951-25, diluted 1:1000) and Myc-tag (Cell Signaling Cat # 2276 S, diluted 1:1000).

### Plasmids

The plasmid $pEZT$-$SPRING_{ecto}$-$His_{10}$ contains, beginning from the N-terminus, an IgK leader secretion tag (METDTLLLWVLLLWVPGSTGD), human SPRING residues 36-205, and ten His residues. The plasmid $pEZT$-$SPRING_{ecto}$-HA is identical to $pEZT$-$SPRING_{ecto}$-$His_{10}$ except that it contains a single HA epitope tag (YPYDVPDYA) instead of the $His_{10}$ tag. $pEZT$-$FLAG$-$S1P_{ecto}$-$His_{10}$ contains, beginning from the N-terminus, an IgK leader secretion tag (METDTLLLWVLLLWVPGSTGD), a single FLAG epitope tag (DYKDDDDK) human S1P residues 23-998, and ten His residues. The plasmid $pEZT$-$S1P_{ecto}$−3xFLAG contains, beginning from the N-terminus, human S1P residues 1-998 followed by a 3xFLAG epitope tag (DYKDDDDKGSDYKDDDDKGSDYKDDDDK). These ORFs were introduced into the pEZT vector backbone[59] using Gibson Assembly methods. Full-length S1P and SPRING constructs were used as PCR templates for the ectodomain constructs and have been described previously[39]. Mutations were introduced using standard site-directed mutagenesis methods and mutagenic primer sequences are provided in Supplementary Table 1. Expression plasmids are available upon request. All plasmid open reading frames were verified by Sanger sequencing.

### Cell culture

HEK293S GnTI⁻ cells were obtained from ATCC (CRL-3022). Cells were thawed in Dulbecco's modified Eagle's medium (DMEM) (high glucose) medium supplemented with 10% v/v fetal calf serum, 100 units/ml penicillin, and 100 mg/ml streptomycin sulfate and passaged in monolayer culture at 37 °C in 5% $CO_2$. After expanding to ten 10 cm dishes, cells were sloughed off and transferred to suspension culture in FreeStyle 293 medium (ThermoFisher) supplemented with 2% v/v FCS, 100 units/ml penicillin, and 100 mg/ml streptomycin sulfate. Suspension cells were grown with orbital shaking at 130 rpm in baffled flasks at 37 °C, and 8% $CO_2$. Cells were passaged to maintain a suspension density within the range of $0.4 \times 10^6$ cells/ml to $2.5 \times 10^6$ cells/ml.

HEK293T cells were obtained from ATCC (ATCC CRL-3216) and HEK293T cells were maintained in monolayer culture at 37 °C and 5%

CO$_2$ in DMEM (high glucose) supplemented with 10% v/v FCS, 100 units/ml penicillin, and 100 mg/ml streptomycin sulfate.

To guard against genomic instability, an aliquot of each cell line was passaged for only 4–6 weeks. Cells were confirmed to be free of mycoplasma using InVivogen MycoStrip kit. Cell lines were not validated further.

## Protein expression and purification

BacMam baculovirus were produced as described previously[59]. Secreted SPRING$_{ecto}$ and S1P$_{ecto}$ proteins were expressed from HEK293sGnTI$^-$ cells by infecting HEK29sGnTI$^-$ cells with BacMam baculovirus using a multiplicity of infection (MOI) of ~3 virions per cell when the cells were at a density of 3 million cells per ml culture. The culture media was supplemented with sodium butyrate to a final concentration of 5 mM and the cells were switched to orbital shaking at 130 rpm at 30 °C, 130 rpm, and 8% CO$_2$. Cells were cultured for 72 h following infection, after which cells were pelleted by centrifugation at 4 °C and 4000 x g for 30 min. Supernatants were harvested and used immediately or else stored at −80 °C until purification. Immediately prior to purification, supernatants were clarified by filtration using 0.2 micron filter units.

To purify His$_{10}$-tagged proteins, supernatants were first passed over NiNTA Superflow resin (Qiagen). Resin was washed with 10–15 CV of buffer consisting of 150 mM NaCl, 50 mM HEPES-NaOH pH 7.5, and 50 mM Imidazole pH 7.5. Proteins were eluted in buffer consisting of 150 mM NaCl, 50 mM HEPES-NaOH pH 7.5, and 300 mM Imidazole pH 7.5. Eluted proteins were further purified by gel over a superdex 200 increase column (Cytiva) equilibrated in buffer consisting of 150 mM NaCl and 50 mM HEPES-NaOH pH 7.5.

To purify S1P$_{ecto}$–3xFLAG, supernatants were first passed over anti-FLAG M2 affinity resin (Sigma) that had been equilibrated with PBS. Resin was washed with 20 CV of buffer consisting of 150 mM NaCl, 50 mM HEPES-NaOH pH 7.5. Proteins were eluted in wash buffer supplemented with 400 μg/mL of FLAG peptide. Eluted proteins were further purified by gel filtration over a superdex 200 increase column (Cytiva) equilibrated in buffer consisting of 150 mM NaCl and 50 mM HEPES-NaOH pH 7.5.

## SPRING$_{ecto}$-S1P$_{ecto}$ co-immunoprecipitation assay

HEK293T cells were set up on day 0 in 6-well plates at a density of 400,000 cells/well. The next day, fresh culture medium was applied, and cells were transfected with 3 μg DNA using X-tremeGENE HP (Roche) according to the instructions of the manufacture. Cells were cultured for 48–72 h, after which the supernatant was collected and clarified by centrifugation at 4 °C and 2000 x g for 10 min. 4% of the supernatants were retained for immunoblot analysis and the remainder were incubated while rotating at 4 °C with 20 μL equilibrated anti-HA resin (Pierce Anti-HA Agarose, ThermoFisher Cat # 26181) for 1 h. Following this incubation, resin was pelleted by centrifugation at 4 °C and 2000 x g for 2 minu and washed by resuspension in 1 mL of buffer consisting of 150 mM NaCl and 50 mM HEPES-NaOH pH 7.5 followed by rotation at 4 °C for 10 min. Resin was washed three times before bound proteins were eluted in 200 μL 200 mM Glycine pH 3.5. The retained input and collected eluates were subjected to SDS-PAGE and immunoblot analysis.

## SPRING$_{ecto}$-S1P$_{ecto}$ competitive pulldown assay

HEK293T cells were set up on day 0 in 10 cm dishes at a density of 1 million cells/dish. The next day, fresh culture medium was applied, and cells were transfected with 6 μg DNA using X-tremeGENE HP (Roche) according to the instructions of the manufacture. Cells were cultured for 72 h, after which the supernatant was collected and clarified by centrifugation at 4 °C and 2000 x g for 10 min. 4% of the supernatants were retained for immunoblot analysis and the remainder were supplemented with 50 mM Imidazole pH 7.5 to reduce non-specific

binding of the SPRING ectodomains to the NiNTA resin. Supernatants were then incubated with 100 μL equilibrated NiNTA resin for 1–2 h at 4 °C while rotating. Following this incubation, resin was pelleted by centrifugation at 4 °C and 2000 x g for 2 min and washed by resuspension in 1 mL of buffer consisting of 150 mM NaCl, 50 mM HEPES-NaOH pH 7.5, and 50 mM Imidazole pH 7.5, followed by rotation at 4 °C for 10 min. Resin was washed three times before bound proteins were eluted in the wash buffer supplemented with Imidazole pH 7.5 to a final concentration of 300 mM. The retained input and collected eluates were subjected to SDS-PAGE and immunoblot analysis.

## S1P$_{ecto}$ peptide cleavage assay

Purified proteins were incubated in a reaction buffer consisting of 150 mM NaCl, 50 mM HEPES pH 7.5, 2 mM CaCl$_2$, 0.01% Tween-20. The reaction was initiated by addition of the self-quenching peptide (Mca-peptide-LysDNP). The reaction was analyzed at 25 °C with excitation at 320 nm and emission at 380 nm using a CLARIOstar plate reader (BMG LABTECH) or else using a Cytation 5 imaging reader (Biotek). Fluorescence was converted to peptide concentration using by digesting the peptide with 100 nM S1P$_{ecto}$-SPRING$_{ecto}$, which showed complete digestion during the duration of the experiment (typically 1 hr). Michaelis-Menten parameters were obtained by calculating initial velocity (defined as the first 5 min of the reaction) using 10 nM enzyme to cleave substrate peptides at a concentration of 10 to 200 μM. All reactions were carried out in triplicate and the similar results were observed with at least two independent purifications of the proteins.

## Mass photometry

Mass photometry was performed by making 100 nM stocks of the protein in PBS, then adding 1.8 μL of the stock to 16.2 μL of 1X PBS that had been positioned on a glass cover slip which had been mounted and focused on a Refeyn TwoMP mass photometer. An interferometric movie of a small portion of the cover slip was taken for 60 s. The movie was converted to a ratiometric format in Discover$^{MP}$, and contrast events were related to masses using a previously obtained standard curve with BSA. The figure was rendered in Discover$^{MP}$. Mass photometry on the sample displayed a single, dominant peak. A Gaussian fit to the peak had a mean of 116 kDa.

## SEC-MALS

A Sephadex 200 Increase 10/300 size-exclusion column was equilibrated with gel filtration buffer. 400 μg of the sample at 2 mg/mL was injected and chromatographed at a flow rate of 0.5 mL/min. Data were recorded on a Shimadzu UV detector, a Wyatt TREOS II light-scattering detector, and a Wyatt Optilab t-REX differential refractive-index detector, which were all calibrated with a BSA standard. SEC-MALS data were analyzed with ASTRA version 7.3.0.11. The molar mass associated with this peak was close to 120,000 g/mol (the weight-average molar mass was 119,800 g/mol). There was a slight slope to the observed mass values.

## Immunoblot

**For ectodomain studies.** Following SDS-PAGE, proteins were transferred to nitrocellulose membranes using the Trans-Blot Turbo Transfer System (BioRad Laboratories, Hercules, CA). Membranes were probed with the primary antibodies and bound antibodies were detected by chemiluminescence (SuperSignal West Pico Chemiluminescent Substrate, Thermo Scientific, Waltham, MA) using a 1:5000 dilution of anti-mouse IgG conjugated to horseradish peroxidase (Jackson ImmunoResearch Laboratories, Inc., West Grove, PA). Membranes were exposed to Blue X-ray Film (Phoenix Research Products, Pleasanton CA).

**For full-length studies.** Total cell lysates for immunoblotting were prepared in RIPA buffer (Boston Biochem), which was supplemented

with 1 mM phenylmethylsulfonyl fluoride (PMSF) (Sigma) and a protease inhibitors cocktail (Roche). Subsequently, lysates were cleared by centrifugation at 4 ˚C at 12,000 x g for 10 min. Media was cleared by a 2-step centrifugation first at 4 °C at 400 x g for 5 min and then at 4 °C at 12,000 x g for 10 min. Samples were separated on NuPAGE Novex 4–12% Bis-Tris gels (ThermoFisher) and transferred to nitrocellulose membranes, blocked in 5% milk (Elk) in PBS supplemented with 0.05% Tween and then probed with primary antibodies as indicated. Secondary HRP-conjugated antibodies (A28177 & A27036, Invitrogen) were used and visualized with chemiluminescence on an IQ800 (GE Healthcare) and quantified using ImageJ2 (Version 2.14.0/1.54 f) with the gel analyzer plugin.

### Cryo-EM grid prep
Quantifoil R 1.2/1.3 300 Mesh, Au grids were glow discharged for 80 s at 30 mA using a Pelco easiglow machine. 3 μL sample at 0.6–0.7 mg/mL was applied at 4 °C in 100% humidity and blotted for 4.5 s followed by freezing in liquid ethane cooled by liquid N2, using a Vitrobot Mark IV.

### Cryo-EM data collection
The S1P$_{ecto}$-SPRING$_{ecto}$ sample was imaged on a Titan Krios G2 operating at 300 kV K3 Summit direct electron detector (Gatan) in the super-resolution CDS counting mode, operated at 300 kV using SerialEM software[60]. The super-resolution pixel size was 0.54 Å, the dose rate was ~8 e-/Å$^2$/sec, and the total accumulated dose was 48 e-/Å$^2$ over 50 frames. A nominal defocus range of −1.0 to −2.0 microns was used, and the energy filter was set to 20 eV.

The S1P$_{ecto}$ sample was imaged on a Titan Krios G3i operating at 300 kV using a Falcon 4i electron detector using SerialEM software[60]. The pixel size was 0.738 Å, the dose rate was ~8 e-/Å$^2$/sec, and the total accumulated dose was 55 e-/ Å$^2$. A nominal defocus range of −0.6 to −1.6 microns was used, and the energy filter was set to 5 eV.

### Cryo-EM image processing
All image processing was carried out using CryoSparc v4.2[61]. For the S1P$_{ecto}$-SPRING$_{ecto}$ sample, 4885 movies were subjected to Patch Motion Correction, where they were Fourier cropped to a pixel size of 1.08 Å, followed by Patch CTF Estimation. Gaussian blob picking was used to generate initial templates and 15.7 million particles were picked. Initial particles were extracted and binned into a pixel size of 2.7 Å. After two rounds of 2D classification, 5.8 million particles were selected and re-extracted and binned to a pixel size of 1.35 Å. Following two rounds of Heterogenous 3D Refinement, ~2 million selected particles were re-extracted in a box of 160 pixels at 1.08 Å/pix. These particles were subjected to Non-Uniform 3D Refinement (NU-3D) to produce a map of 2.86 Å but which had relatively poor features. A mask was generated covering a lobe that would be revealed to be the bound SPRING$_{ecto}$ and this mask was used for focused 3D classification without alignment. Two classes containing 452,453 particles were selected for containing strong features for SPRING$_{ecto}$. These particles were re-extracted in a box of 300 pixels at 1.08 Å/pix, which yielded a map of 2.4 Å. These particles were subjected to local motion correction and extracted a box of 600 pixels with the super-resolution pixel size of 0.54 Å. NU-3D with Global CTF Refinement and per-particle defocus optimization yielded a map of 2.3 Å. For illustration purposes, the final map is sharpened with a B-factor of −70 unless indicated.

For the S1P$_{ecto}$ sample, 6139 movies in .eer format were subdivided into 50 frames and import without upsampling the 4 K grid. Following Patch Motion Correction and Patch CTF Estimation, 5890 curated micrographs were selected for further analysis. Gaussian blob picking was used to generate initial templates and 7.8 million particles were picked using template picking. Particles were extracted and binned to a pixel size of 2.5 Å. Two rounds of 2D classification yielded 1.9 million particles, which were subjected to round of Heterogenous 3D Refinement against three ab-initio classes to yield 965,595 particles that were

re-extracted and binned into a pixel size of 0.92 Å. Another round of Heterogenous 3D Refinement yielded a subset of 724,742 particles. Non-Uniform 3D Refinement of these particles produced a map with a nominal resolution of 2.27 Å but which was of relatively poor quality. These particles were re-extracted in a box of 400 unbinned pixels, an ab-initio map was generated, and subsequent NU-3D refinements used an initial low pass filter of 10 Å to obtain maps of higher quality for this small protein. The particles were subjected to local motion correction and a round of 3D classification without alignment was carried out to remove bad particles. A final set of 659,523 particles were used to generate the final map with a resolution of 2.27 Å using GSFSC and 3DFSC. For illustrations, the final map was sharpened with a B-factor of −62 unless indicated.

### Model building
Initial models were generated from the Alphafold 2 database obtained via UniProt. Starting models were subjected to rounds of manual building in Coot[62] and ISOLDE[63] and real-space refinement in Phenix[64]. Refinement statistics are provided in Table 1.

### Quantification and statistical analysis
Cryo-EM data collection, analysis, and refinement statistics are shown in Table 1. All experiments were reproduced in at least three independent experiments. Purified protein assays were carried out with samples from at least two independent purifications. Sequence alignment figures were generated by alignment with Clustal server and visualized with ESPript 3.0 server[65]. For western blot data, statistical significance was tested using one-way ANOVA with Holm-Sidak post hoc analysis. Prism v10.1.0 software was used for statistical analyses and $p < 0.05$ was considered significant. $p$-values are indicated by asterisk: ***$p < 0.001$, ****$p < 0.0001$.

### Reporting summary
Further information on research design is available in the Nature Portfolio Reporting Summary linked to this article.

## Data availability
The atomic models generated in this study are deposited in the Protein Data Bank (PDB) with codes PDB 8UW8 (S1P$_{ecto}$-SPRING$_{ecto}$) and PDB 8UWC (S1P$_{ecto}$) and the cryo-EM maps are deposited in the Electron Microscopy Data Bank (EMDB) with the accession codes: EMD-42639 (S1P$_{ecto}$-SPRING$_{ecto}$) and EMD-42661 (S1P$_{ecto}$). The raw movies have been deposited to EMPIAR with the accession codes: EMPIAR-11928 (S1P$_{ecto}$-SPRING$_{ecto}$) and EMPIAR-11931 (S1P$_{ecto}$). Atomic models generated previously that were used in this study are PDB 4Z2A and PDB 2PMW. Source data are provided with this paper. All other data are available from the corresponding authors upon request. Source data are provided with this paper.

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

## Acknowledgements
This project was supported in part by R00GM141261 (to DLK). HH was supported through the Dr. Emmett J. Conrad Leadership Program. N.Z. is an Established Investigator of the Dutch Heart Foundation (2013T111) and is supported by a Vici grant from the Netherlands Organization for Scientific Research (NWO; 016.176.643) and an NWO ENW grant (M.22.034; GENESIS). All cryo-EM data was collected at the UT Southwestern Cryo-Electron Microscopy Facility (CEMF). We thank Dan Stoddard, Ph.D., and the CEMF staff for assistance with cryo-EM data collection. The CEMF is supported by a core facilities award from the Cancer Prevention & Research Institute of Texas (CPRIT RP220582). We thank Chad Brautigam and Shih-Chia Tso for carrying out SEC-MALS and Mass Photometry experiments. We thank members of the Zelcer and Kober labs and Irith Koster for their critical comments and suggestions on this study.

## Author contributions
S.H. conceptualized the study, developed and carried out western blot experiments and revised and edited the paper. V.D. developed the protease assay, determined the enzyme kinetics, and developed and carried out the competitive coIP assay. H.H. carried out the protein purifications for cryo-EM grid preps, carried out biochemical characterization of the protein samples, and conducted initial coIP assays. S.B. carried out protease assays. J.K. carried out western blot experiments. B.B. carried out with molecular biology and protein expression experiments. N.Z. conceived the project, supervised research, and wrote the paper. D.L.K. conceived the project, supervised research, did the cryo-EM analysis, carried out biochemical assays, and wrote the paper.

## Competing interests
The authors declare no competing interests.
