## [Peer Review File · Nature Communications]

SPRING licenses S1P-mediated cleavage of SREBP2 by displacing an inhibitory pro-domainREVIEWER COMMENTS

Reviewer #1 (Remarks to the Author):

In this manuscript, Hendrix et al describe the biochemical characterizations of the activation of site 1 protease (S1P) by SPRING and cryoEM structures of S1P in the presence and absence of SPRING. S1P plays the key role in the activation of sterol regulatory element binding protein via initiating the first of two proteolytic processes. The overall quality of work is high and this study presents a major advance in how S1P is kept in the inactive state and how SPRING binding activates S1P. I have the following suggestions to improve this manuscript.

1. Michaelis-Menten kinetics analysis shown in Figure 2 is measured using SPRING-bound S1P. Authors refer others' work on the similar analysis, which is done using different peptide substrate. For a fair comparison, it would be nice to have the enzyme kinetic parameters of unstimulated S1P using the peptide substrate described in this manuscript.
2. Authors only have relatively limited descriptions of S1P structure. For the structural audiences, authors should expand their description in such analysis. This can include (but not be limited to) domain-domain interaction between peptidase, P-like domain and GIFT domain, negative charge groove formed between peptidase, P-like and GIFT domain in term of substrate selectivity (Figure 3e).
3. The map quality for SPRING appears to be not as good as S1P. Authors should provide a more realistic demonstration in the figure using heat map or B factor, e.g., Figure 3 or additional supplemental figure. Authors should also describe whether it make sense for the missing region in SPRING (e.g., lack of contact) or it is due to cryo-freezing of SPRING-bound S1P and partial denaturation.
4. Authors put forth two mechanisms for the activation of S1P by SPRING, one by removal of B-site peptide in the active site and the other by flexible oxyanion hole residue N338. For the second mechanism, authors should elaborate/speculate the mechanism that leads to the stabilization of N338. Furthermore, authors may consider to briefly summarize the field for the mechanism of protease activation and whether the mechanisms described here is the common mechanism.

Reviewer #2 (Remarks to the Author)

In this monumental work, the authors concentrate on the role of SPRING in S1P activation. They approached it from a biochemical and structural point of view. Their conclusions suggest that SPRING is necessary for the S1P activation and removal of the N-terminal A segment of the prodomain of S1P.

The following suggestions are intended to enhance the quality and readability of this manuscript:

1. Please correct all the figures that depict S1P to replace the TM by a TM-CT as the cytosolic tail is a major part of the protein not to be confused with the transmembrane domain, e.g., Figs. 1a, 2c, ext-Fig 1a.
2. The A domain of S1P is structured and highly conserved between human and drosophila (please refer to Ref 39). What is the situation with SPRING, it looks like the S1P binding residues of SPRING are identical between human and drosophila.

```

sp|Q9H741|SPRING_HUMAN      MVNLAAMVWRLLRKRWV LALVFLS L LVYFLSSTFKQEEERAVRDRNLLQVVDHNNQIPWVKVQFN64
tr|A0A6P4EER4|A0A6P4EER4_DRORH  ---MWPALVLPRLRLRRRIYIVLVLLV LVYVGFGLFDGDSFSGSLHY-DEYV VQRTRPLLW TQQLL60

Q9H741:Chain

sp|Q9H741|SPRING_HUMAN      L-----GNSSRPSNQCRNSIQGKHLITDELGYVCEK KDLVNGCCNVNVPSTKQYCCDGCWP-121
tr|A0A6P4EER4|A0A6P4EER4_DRORH  APEEQLNRTRGDPEARCRNSVQGRRLADERGFVCRREEEVLISGCCNPELPGIGYVSCRTCNTT124

Q9H741:Chain

sp|Q9H741|SPRING_HUMAN      NGCCSA YEYCVS CCLQPNKQLLLERFLNRAAVAFQNLFMAVEDHFFELCLAKCRTSSQSVQHENT185
tr|A0A6P4EER4|A0A6P4EER4_DRORH  THCCGVYEYCVS CCLHFGQQPLLERVLQAPN-TPKYIFASVSDHFFELCLVKCRTNSHSHVEHENK187

Q9H741:Chain

sp|Q9H741|SPRING_HUMAN      YRDP IAKYCYGESPPPEL FPA-----
tr|A0A6P4EER4|A0A6P4EER4_DRORH  YRDPAAKHCHYGLFTEAHESQRNVAPQASRS

Q9H741:Chain

```

Thus, in both species are conserved critical residues SPRING R95, R174, V180.

On the S1P side the critical three residues are also identical : P317, N328 , W556

```

sp|Q14703|MBTP1_HUMAN      MKLVNIWLLLLVLLCGK KHLGDRLEKKSFEKAPCPGCSHLTLKVEFSSTVVEY EYIVAFNGYF64
tr|A0A034WJ4|A0A034WJ4_BACDO  MLFLVTRII L FSS IYTSN KHL-----KIYGVQCTH-AEEVRFESKILENEYIVQYRHY53

Q14703:Signal

sp|Q14703|MBTP1_HUMAN      TAKARNSFISSALKSSEVDNWR IIPRNPSSDYPSDFEV I QIKEK--QKAGL L TLE DHPN I KRV126
tr|A0A034WJ4|A0A034WJ4_BACDO  FPARRKYLNAAFRRANILEWNLVERLNAALNYPDFDLIKLEYSDLA I TQ I L QLELHPLVKV117

Q14703:Signal

sp|Q14703|MBTP1_HUMAN      TPQRKVF RSLKYAESDPTVPCNETRWSQKWQSSRPLRRASLSLGS GFWHATGRHSSRRL LRAIF190
tr|A0A034WJ4|A0A034WJ4_BACDO  SPQRSVERILN YNQTR-----ERIGRQILRAIF145

Q14703:Signal

sp|Q14703|MBTP1_HUMAN      RQVAQTLOADV L WQMGYTGANVRVAVFD TGLSEKHPHFKNVKERTNWTNERTLDDGLGHGTFVA254
tr|A0A034WJ4|A0A034WJ4_BACDO  TQITTMLQANVLWGMGI TGA GVKVAVFD TGLAKSHPFERRVKERTNWTNEKSLDDGVSHGTFVA209

Q14703:Signal

sp|Q14703|MBTP1_HUMAN      GVIASMR ECGGFAPDAELHI FRVFTNNOVSYTSWFLDAFN YAI LKKIDVNLNLSIGGPDFMDHFF318
tr|A0A034WJ4|A0A034WJ4_BACDO  GVIASAK ECLGLAPDVELHIYRVFTNSQVSYTSWFLDAFN YAI FKKIKVNLNLSIGGPDFLDRFF273

Q14703:Signal

sp|Q14703|MBTP1_HUMAN      VDKVWELTANNVIMVSAIGNDGLYGT LNNFADQMDVIGVGGIDFEDNIARFSSRGMTTWELFG382
tr|A0A034WJ4|A0A034WJ4_BACDO  VDKVLELSANKVIMISAIGNDGLYGT LNNFGDQSDVIGVGGINFEHKVAKFSSRGMTTWELFW337

Q14703:Signal

sp|Q14703|MBTP1_HUMAN      GYGRMKPDI VTYGAGV RGS GVKGGCRA LSGTSVSPVAVAGAVTLLVST-VQKRELVN PASMKG A445
tr|A0A034WJ4|A0A034WJ4_BACDO  GYGR LKPDIVTFG SQV KGSNVHGGCRSLSGTSVSPVAVAGAVL IASGALS KLN L I N P S S M K G I401

Q14703:Signal

sp|Q14703|MBTP1_HUMAN      L I A S A R R L P G V N M F E O G H G K L D L L R A Y Q I L N S Y K P Q A S L S P S Y I D L T E C P Y M W P Y C S Q P I Y Y G509
tr|A0A034WJ4|A0A034WJ4_BACDO  L M E G A E R L P D N M F E O G H G K L D I L K S M Q L L T Y E P K I T L S P E Y L D T E - S Y M W P Y S S Q P I Y Y G S464

Q14703:Signal

sp|Q14703|MBTP1_HUMAN      M P T V V N V T I L N G M G V T G R I V D K P D W Q P Y L P Q N G D N I E V A F S Y S S V L A P W S G Y L A I S I S V T K K A A573
tr|A0A034WJ4|A0A034WJ4_BACDO  M P I I A N V T I L N G I A V T G V I K D A P L W H P H T D K H G N V L N V A V S Y S T N L A P W S G W M S V H I V V N E K G D528

Q14703:Signal

sp|Q14703|MBTP1_HUMAN      S W E G I A Q G H V M I T V A S P A E T E S K N G A E Q T S T V K L P I K V K I I P T P P R S K R V L W D Q Y H N L R Y P P G Y637
tr|A0A034WJ4|A0A034WJ4_BACDO  N F K G V A E G H I S L E I E C L Q E Q Q - - E S K I V T K I D F P L K V T V I P K P P R N K R I I W D Q Y H S L K Y P P G Y589

```

3. On page 404, the authors suggest that S1P and SPRING interact in the Golgi and not the ER. However, evidence from arenavirus activation by S1P suggests that in some cases processing can occur in the ER/cisGolgi, especially for best S1P substrates. Thus, using soluble S1P-KDEL (S1P_{ecto}-KDEL) it was shown that viral glycoprotein processing occurs quite efficiently for LASV (RRL↓), but not for LCMV (RRLA↓) (see PMID: 22357276, Fig. 3C). Thus, for some more efficiently cleaved substrates (RRL↓) activity of S1P must exist in the ER/cis Golgi. It is suggested that the addition of a KDEL sequence at the C-terminus of SPRING_{ecto} (SPRING_{ecto}-KDEL) may help resolve this issue for SREBP2, which likely has a less efficiently cleaved S1P site (RSVL) as also applicable to LCMV (RRLA↓) (see review PMID: 34202098) .
4. Can the authors speculate on the possible role of the variant (Q55R-SPRING) in apoA1 and HDL regulation (ref 34)? What is the effect of this mutation of SPRING activity on SREBP or S1P based on the present structural data, and how does that compare to the speculation on the consequence of Q55R in the previous publication based solely on Alpha2 fold?
5. One major problem this referee has, is how does the Ser₄₁₄ active site mutant S1P_{ecto}-S414A exit the ER (as it is usually retained in the ER as proS1P) and be secreted? Is the secretion dependent on SPRING? This is highly unusual, as the only zymogen that can exit the ER is proPCSK2-bound to its chaperone 7B2. Can this be the case here for proS1P-SPRING? This would further support for the notion that S1P binds SPRING very early in the secretory pathway (e.g., ER), as is the case for 7B2 to proPCSK2. Can the authors address this question and possibly show that S1P_{ecto}-S414A cannot exit cells lacking SPRING and that it can if SPRING is re-expressed in these cells. They claim that they can purify proS1P_{ecto}-S414A from HEK293 cells lacking SPRING? This is really strange and requires further clarification.
6. The authors claim that the A segment of S1P competes with SPRING for the binding to S1P. Can the authors speculate where the A segment binds in S1P, as opposed to SPRING? The only 5 aa homology of sequence between the A segment and SPRING this referee found is:

```

- - - - - Y F L S S T F K Q E - - - - - 38
spjQ9H741|SPRING_HUMAN
spjQ14703|MBTP1_HUMAN E Y I V A F N G Y F T A K A R N S F I S S A L K S S E V D N W R I I P R N N P S S D Y P S D F E V I Q I K E K Q K A G L L T L E 118
Q9H741:Chain

```

The rationale of this conserved FxSSxxK in the A segment and SPRING is not clear.

7. The need for SPRING in activation of S1P suggests that it may be critical in other S1P-cleavages distinct from SREBP, e.g., viral glycoprotein. The authors may wish to expand their discussion to speculate on the other critical functions of SPRING , e.g., in viral infections.

Reviewer #3 (Remarks to the Author):

This study using protein-pulldowns, enzymatic assays and structural biology show that SPRING binds to S1P, dislodges the inhibitory A-domain which results in an active form of S1P that can efficiently cleave a fluorogenic peptide substrate. This study is well designed, clearly written and the conclusion is supported by the data. S1P plays an important role in regulating transcription factors, hormones, and enzymes. Therefore this study provides us with key insights into the regulation of S1P activity by SPRING. I have no additional experiments to suggest, but have a few comments that will likely improve the manuscript.

Comments:

Line 114: "Co-expressing a P4-mutant SPRINGectoR45A 114 with S1Pecto resulted in a single band for SPRING".

Please add a sentence and a reference before or after this statement to indicate why an R to A mutation in the P4 position of the cleavage site would prevent cleavage. This will also be important later on as the same amino acid is changed in the fluorogenic substrate. Why is the P4 so important for S1P?

Line 144: We have been unable to detect whether the cleaved B domain is bound to either S1Pecto or SPRINGecto-S1Pecto owing to its small size (~3-5 kDa).

It would be valuable to the reader to re-draw Figure 1a showing that the B domain (and the signal peptide) is considerably smaller than the A domain. Based on the current graphic, I assumed that they were approximately the same size.

Figure 1i: The SPRINGecto-HA mixed with the A form (Lane 6) appears to be a higher mass (~30 kDa) than the other two lanes. This is most evident in the eluate. Can you address why this may be occurring??

Line 183: Change PF429242 to PF-42924 (term used in the methods section) and add a reference as to the discovery of this inhibitor

Line 309: Even though the GIFT domain cannot be resolved, it would be helpful for the reader if a supplementary figure could be made that contains side-by-side structures of S1P/Spring (e.g Fig 3a) and S1P with A domain (e.g Fig 5a) in the exact same orientation so that it is easier to visualize how SPRING and the A domain have overlapping binding sites. I found myself flipping back and forth between Figure 3 and Figure 5 to compare them.

Reviewer #1 (Remarks to the Author)

In this manuscript, Hendrix et al describe the biochemical characterizations of the activation of site 1 protease (S1P) by SPRING and cryoEM structures of S1P in the presence and absence of SPRING. S1P plays the key role in the activation of sterol regulatory element binding protein via initiating the first of two proteolytic processes. The overall quality of work is high and this study presents a major advance in how S1P is kept in the inactive state and how SPRING binding activates S1P. I have the following suggestions to improve this manuscript.

We are grateful for the reviewer's favorable summary of our work and have adopted their suggestions to improve the manuscript.

1. Michaelis-Menten kinetics analysis shown in Figure 2 is measured using SPRING-bound S1P. Authors refer others' work on the similar analysis, which is done using different peptide substrate. For a fair comparison, it would be nice to have the enzyme kinetic parameters of unstimulated S1P using the peptide substrate described in this manuscript.

We appreciate the opportunity to expand on this important observation. A key finding from our original manuscript was that S1P without SPRING lacked detectable activity (**Fig 2b**) whereas S1P in complex with SPRING has strong activity. As such, we could not carry out a comparable analysis to measure the Michaelis-Menten parameters for S1P lacking SPRING. However, we have taken the opportunity to update these experiments in two important ways:

- 1) We carried out additional enzyme analyses where we extended the substrate concentration to 200 μM (from 100 μM) to obtain results closer to saturation. This resulted in a more accurate estimate of K_m and V_{max} (we also see some decrease in V_{max} because of the increased DMSO concentration from the peptide stock).
- 2) We further probed the activity of S1P without SPRING. The initial report characterizing purified S1P showed activity against a fluorescent version of S1P B-site peptide (VFRSLK), which resulted in the K_m/K_{cat} measurements. In contrast, there was little to no fluorescent activity with a SREBP2 peptide of matched length (SGRSVL). Cleavage of a 16-mer SREBP2 peptide was detected by HPLC following a 4 hr incubation of peptide with ~ 600 nM enzyme at 37°C (PMID: 10428865). We noted in the original manuscript that our SREBP2 K_m is in the same regime as reported previously (~ 100 μM) while the V_{max} of our S1P-SPRING complex against SREBP2 is $\sim 1000x$ faster than proteolysis data using the optimal S1P B-site peptide.

We considered several possible explanations for the lack of activity by S1P in our study, which are also mentioned in the "Discussion" section.

- **Possibility 1** is that our experiment comparing FLAG-S1P_{ecto}-His₁₀ to the S1P_{ecto}-3xFLAG/SPRING_{ecto}-His₁₀ complex reported an artifact resulting from the different tags on S1P_{ecto}. To test this, we used anti-FLAG affinity chromatography to purify S1P_{ecto}-3xFLAG and compared it to S1P_{ecto}-3xFLAG/SPRING_{ecto}-His₁₀. In this experiment, the S1P constructs are exactly matched. We see that the S1P_{ecto}-3xFLAG is still inactive without SPRING and we have added this data to the manuscript as **Sup. Fig 1j-k**. This confirms the loss of activity was not due to the different purification tags.

- **Possibility 2** is that S1P alone has only some weak activity that is clear with longer incubation at higher temperatures. It is likely that, with enough time, the A-domain might spontaneously dissociate and allow sporadic activity at some low rate. However, we suspect this is not a physiologically-relevant amount of activity.

- **Possibility 3** is that the earlier reported experiments purified some endogenous SPRING in complex with the exogenous S1P and that this "contaminating" complex was providing the measured activity. This is plausible since SPRING can be cleaved by S1P and secreted into the media (e.g. **Fig. 5**). Our modern expression and purification methods may be able to purify larger amounts of S1P relative to any co-purified endogenous SPRING and therefore the S1P would appear less active than previous

reports with purified S1P. We cannot test this hypothesis directly, but it is the explanation we think is the most likely.

2. Authors only have relatively limited descriptions of S1P structure. For the structural audiences, authors should expand their description in such analysis. This can include (but not be limited to) domain-domain interaction between peptidase, P-like domain and GIFT domain, negative charge groove formed between peptidase, P-like and GIFT domain in term of substrate selectivity (Figure 3e).

We thank the reviewer for the opportunity to expand the description of S1P±SPRING and we hope it is of interest to the audience. In the revised **Figure 3** and **Figure 6** (previously Fig. 5) we provide additional structural analysis as requested. We have also added several additional panels in the supplementary data supporting the expanded descriptions. Finally, we expanded the speculative discussion of substrate specificity by comparing the structures of S1P to S1P-SPRING (**new Fig. 7**)

3. The map quality for SPRING appears to be not as good as S1P. Authors should provide a more realistic demonstration in the figure using heat map or B factor, e.g., Figure 3 or additional supplemental figure. Authors should also describe whether it make sense for the missing region in SPRING (e.g., lack of contact) or it is due to cryo-freezing of SPRING-bound S1P and partial denaturation.

We agree that putting the model in context of the experimental data (the cryo-EM map) is important and we thank the reviewer for the suggestion about how to better communicate this to readers. The density for SPRING, while not uninterpretable, certainly suffers from more heterogeneity than S1P. We have added new analysis (**Supplemental Fig. 3g**) showing the S1P-SPRING structure colored by B factor. SPRING residues interacting with S1P show similar B-factors as neighboring S1P residues while the loopier regions to the exterior are more poorly resolved. We also took this opportunity to revise **Fig 3a** and present the map and model fits more clearly.

The unresolved portions of SPRING likely reflect lack of contacts with S1P. We have added a new panel in **Fig. 3i** showing the missing regions of SPRING in the context of the built model to highlight their separation from S1P. There may also be some heterogeneity in SPRING itself. The lack of major secondary structure elements means the protein is dependent on its complex disulfide bond network for structure. We have expanded our discussion of these points in the text describing the structure of SPRING (related to point #2 above).

4. Authors put forth two mechanisms for the activation of S1P by SPRING, one by removal of B-site peptide in the active site and the other by flexible oxyanion hole residue N338. For the second mechanism, authors should elaborate/speculate the mechanism that leads to the stabilization of N338. Furthermore, authors may consider to briefly summarize the field for the mechanism of protease activation and whether the mechanisms described here is the common mechanism.

We have taken this opportunity to elaborate on how the GIFT domain stabilizes the loop containing oxyanion hole residue N338 (**New Fig. 7a**). In addition, we added text to the introduction summarizing the catalysis mechanism of serine proteases and the role of the oxyanion hole residue.

To our knowledge, the displacement of the pro-domain by another protein is a mechanism so far unique to S1P. However, we suspect the regulated release of these pro-domains may be an underappreciated mechanism for other pro-protein proteases.

Reviewer #2 (Remarks to the Author)

In this monumental work, where the authors concentrate on the role of SPRING in S1P activation. They approached it from a biochemical and structural point of view. Their conclusions suggest that SPRING is necessary for the S1P activation and removal of the N-terminal A segment of the prodomain of S1P.

The following suggestions are intended to enhance the quality and readability of this manuscript:

We thank the reviewer for the very positive assessment of our work and have taken their suggestions as detailed below:

1. Please correct all the figures that depict S1P to replace the TM by a TM-CT as the cytosolic tail is a major part of the protein not to be confused with the transmembrane domain, e.g., Figs. 1a, 2c, ext-Fig 1a.

The reviewer's correction is well-taken. Human S1P has a cytoplasmic tail containing residues ~1025-1052. We have added the descriptor TM-CT to **Fig. 1a** and **Fig. 3c**. (Please note that **Supplemental Fig. 1a** described the secreted constructs lacking a C-term transmembrane helix and cytoplasmic tail). We have added short lines to the S1P schematics to represent the CT (e.g. **Figs 1i, 5a, and 8c-d**). We have also made note of the CT in the text describing the topology of S1P in the introduction. Similarly, SPRING contains a ~17 amino acid N-terminal tail predicted to project into the cytoplasm. We have added analogous corrections to figures depicting full-length SPRING. Finally, to avoid confusion, we have added a subscript "FL" to denote experiments using the full-length S1P and SPRING constructs (e.g. **Fig. 5**).

2. The A domain of S1P is structured and highly conserved between human and drosophila (please refer to Ref 39). What is the situation with SPRING, it looks like the S1P binding residues of SPRING are identical between human and drosophila.

SPRING is highly conserved, particularly within mammals, and is well-conserved between human and drosophila. The residues we mutated in SPRING that disrupt the interaction with S1P (R95, R174, V180) are all conserved in drosophila. Please see **Reviewer Figure 1** below. We also note that the 14 cysteine residues that are predicted by AF2 to be in disulfide bonds are invariant while the cysteine predicted to be unpaired is not conserved in drosophila. This suggests the SPRING orthologs share a common fold that has evolved to bind S1P.

Review Figure 1

3. On page 404, the authors suggest that S1P and SPRING interact in the Golgi and not the ER. However, evidence from arenavirus activation by S1P suggests that in some cases processing can occur in the ER/cisGolgi, especially for best S1P substrates. Thus, using soluble S1P-KDEL (S1Pecto-KDEL) it was shown that viral glycoprotein processing occurs quite efficiently for LASV (RRL↓), but not for LCMV (RRLA↓) (see PMID: 22357276, Fig. 3C). Thus, for some more efficiently cleaved substrates (RRL↓) activity of S1P must exist in the ER/cis Golgi. It is suggested that the addition of a KDEL sequence at the C-terminus of SPRINGecto (SPRINGecto-KDEL) may help resolve this issue for SREBP2, which likely has a less efficiently cleaved S1P site (RSVL) as also applicable to LCMV (RRLA↓) (see review PMID: 34202098).

This is an interesting question. We have previously shown that a version of SPRING harboring a KDEL ER retrieval sequence cannot rescue the SREBP response in SPRING-deficient cells (PMID: 32111832) and another group reported a similar observation (PMID: 32694732). This suggests that, at least for SREBP cleavage, SPRING must reside in the Golgi for some amount of time to act on S1P. As the reviewer notes, SREBP2 may be an inefficient S1P substrate. It is possible that SPRING will not be necessary for cleavage of every S1P substrate. We are excited to explore the spatial regulation of the SPRING-S1P interaction, which will require developing tools to study these proteins in endogenous conditions. We have added the SPRING KDEL point to the discussion and cited PMID 34202098 in the discussion of the role of S1P in viral maturation.

4. Can the authors speculate on the possible role of the variant (Q55R-SPRING) in apoA1 and HDL regulation (ref 34)? What is the effect of this mutation of SPRING activity on SREBP or S1P based on the present structural data, and how does that compare to the speculation on the consequence of Q55R in the previous publication based solely on Alpha2 fold?

Unfortunately, the Q55 residue was not observed in our structure (please see new Fig. 3i), so we do not have any insights beyond what we speculated previously. The Q55R variant may change interactions with nearby residues in the linker between the N-terminal transmembrane helix and the S1P-interacting ectodomain (speculated in PMID: 37626055). Here, we can only add that our experimental data show the alphafold2 prediction for the ectodomain of SPRING was very accurate.

5. One major problem this referee has, is how does the Ser414 active site mutant S1Pecto-S414A exit the ER (as it is usually retained in the ER as proS1P) and be secreted? Is the secretion dependent on SPRING? This is highly unusual, as the only zymogen that can exit the ER is proPCSK2-bound to its chaperone 7B2. Can this be the case here for proS1P-SPRING? This would further support the notion that S1P binds SPRING very early in the secretory pathway (e.g., ER), as is the case for 7B2 to proPCSK2. Can the authors address this question and possibly show that S1Pecto-S414A cannot exit cells lacking SPRING and that it can if SPRING is re-expressed in these cells. They claim that they can purify proS1Pecto-S414A from HEK293 cells lacking SPRING? This is really strange and requires further clarification.

We were also surprised by the secretion of S1P_{ecto}^{S414A} from our HEK 293 cells. Likewise, S1P_{ecto}^{N515Q}, which is deficient in N-linked glycosylation, also secretes without maturing to the C-form (Supplemental Fig. 3j). These secretions did not require co-expression with exogenous SPRING, but we have not tested whether the secretion of S1P_{ecto}^{S414A} is dependent on endogenous SPRING present in these cells. We doubt this is the case, as the ectodomains of S1P^{S414A} and SPRING do not appear to interact (Fig. 1i and Fig 4d of the revised manuscript). As the reviewer notes, this secretion is at odds with the established literature showing autocatalysis is required for trafficking out of the ER (Elegoz et al. 2002. Ref 15 in the original manuscript). We favor the trivial explanation that HEK cells are good at secreting overexpressed, immature, and/or misfolded proteins. For example, HEK 293 cells were shown to secrete proteins containing improper intermolecular disulfide bonds (ex: PMID 27995897, see Fig. 2 of that paper). In the original manuscript, we mentioned as “data not shown” that a S1P_{ecto} construct harboring mutations to the B and B' sites (R130E/R134E) could not be secreted from HEK cells. We can speculate that S1P S414A is relatively more stable than the R130E/R134E mutant. The R130E/R134E

mutant would not be cleaved and the mutated residues would electrostatically unfavored from binding the electronegative pockets that recognize the Arg residues (e.g. **Fig. 6c,e**). This may present a more significant impediment to folding than the S414A mutation. We include this data below as **Reviewer Figure 2**. We have added a note on this point to the results and the discussion highlighting this as a limitation to the current study:

Results:

Because a full-length version of S1PS414A, which cannot mature, has been previously shown reside in the ER²³, we suspect the secretion of S1P_{ecto}^{S414A} is an artifact of the strong over-expression regime in HEK cells.

Discussion:

Finally, while our experiments took advantage of the ability of HEK 293 cells to produce soluble, secreted S1P_{ecto}^{S414A} for biochemical studies, it is well-established that immature full-length S1P is retained in the ER²³.

(Ref 23): Elagoz, A., Benjannet, S., Mammabassi, A., Wickham, L. & Seidah, N.G. Biosynthesis and Cellular Trafficking of the Convertase SKI-1/S1P: ECTODOMAIN SHEDDING REQUIRES SKI-1 ACTIVITY*. *Journal of Biological Chemistry* 277, 11265-11275 (2002).

Reviewer Figure 2

6. The authors claim that the A segment of S1P competes with SPRING for the binding to S1P. Can the authors speculate where the A segment binds in S1P, as opposed to SPRING? The only 5 aa homology of sequence between the A segment and SPRING this referee found is FxSSxxK in the A segment and SPRING.

We thank the reviewer for the opportunity to clarify this important point. Our structural analysis of S1P found that the A domain and SPRING bind the C domain of S1P with overlapping epitopes. We apologize that this was not clear in the first manuscript, and we have extensively revised those panels to make it clearer. Please see the new **Figs. 6f, 6g, 6h, and Supplemental Fig. 5b**. The overlapping nature of their epitopes is also presented in **Supplemental Fig. 5d-g and Supplemental Fig. 6**.

We agree it is remarkable that SPRING and S1P(A) compete for the same surface on S1P(C) despite no structural or sequence similarity. We have added two new supplemental panels showing that the few residues between SPRING and the A-domain that are aligned by the clustal server do not play similar roles structurally (**Supplemental Fig. 5b-c**). For example, we highlight a YEY motif present in both proteins. In SPRING, this motif is in a helix away from S1P. In S1P-A, this motif is at the end of a beta strand. We conclude any weak alignment is spurious enough to differ depending on the algorithm (i.e. clustal web server did not identify the FxSSxxK motif as a match) and any minor sequence convergences are not predictive of their structural roles.

7. The need for SPRING in activation of S1P suggests that it may be critical in other S1P cleavages distinct from SREBP, e.g., viral glycoprotein. The authors may wish to expand their discussion to speculate on the other critical functions of SPRING, e.g., in viral infections.

We have added the following text to the discussion:

“Additionally, it will be interesting to test whether SPRING is required for the S1P-mediated maturation of viral glycoproteins from Arenaviruses and other S1P substrates^{1,11,57,58}. Such studies may provide insights as to the cellular compartment in which SPRING and S1P interact.”

Reviewer #3 (Remarks to the Author):

This study using protein-pulldowns, enzymatic assays and structural biology show that SPRING binds to S1P, dislodges the inhibitory A-domain which results in an active form of S1P that can efficiently cleave a fluorogenic peptide substrate. This study is well designed, clearly written and the conclusion is supported by the data. S1P plays an important role in regulating transcription factors, hormones, and enzymes. Therefore this study provides us with key insights into the regulation of S1P activity by SPRING. I have no additional experiments to suggest, but have a few comments that will likely improve the manuscript.

We thank the reviewer for their positive assessment of our work and have taken their suggestions as detailed below:

Comments:

Line 114: “Co-expressing a P4-mutant SPRINGectoR45A 114 with S1Pecto resulted in a single band for SPRING”.

Please add a sentence and a reference before or after this statement to indicate why an R to A mutation in the P4 position of the cleavage site would prevent cleavage. This will also be important later on as the same amino acid is changed in the fluorogenic substrate. Why is the P4 so important for S1P?

We thank the reviewer for the opportunity to expand on this important characterization of S1P and its substrate scope. We have addressed this in several places, notably:

- 1) We have added a section in the first paragraph of the introduction describing the S1P recognition motif in more detail and referenced literature showing the requirement of the P4 Arg residue *in vitro*.
- 2) We added an additional reference for the importance of this motif when discussing the experiment with the mutant peptide.
- 3) We expanded the discussion of the S1P structure and revised a panel (**Fig. 6e**) to highlight the recognition of the B-site's P4 Arg by S1P when it is bound to the A domain.

Line 144: We have been unable to detect whether the cleaved B domain is bound to either S1Pecto or SPRINGecto-S1Pecto owing to its small size (~3-5 kDa).

It would be valuable to the reader to re-draw Figure 1a showing that the B domain (and the signal peptide) is considerably smaller than the A domain. Based on the current graphic, I assumed that they were approximately the same size.

We thank the reviewer for this suggestion and have modified the figures as suggested: we doubled the size of the A domain in **Fig. 1a** and changed the shape and size of the B pro-domain in **Fig. 1b, Fig 1i, Fig 4a, Fig. 5A, Fig 8c, and Fig 8d**.

Figure 1i: The SPRINGecto-HA mixed with the A form (Lane 6) appears to be a higher mass (~30 kDa) than the other two lanes. This is most evident in the eluate. Can you address why this may be occurring??

We suspect the higher MW of SPRING co-secreted with the A-form of S1P is because the A form of S1P does not cleave SPRING. We have added text to the results clarifying this difference.

Line 183: Change PF429242 to PF-42924 (term used in the methods section) and add a reference as to the discovery of this inhibitor

We apologize for the omission of this important citation (Hay et al. 2007), which has been added. We have made the correction of PF429242 to PF-42924 in the Results and in the figure legends.

Line 309: Even though the GIFT domain cannot be resolved, it would be helpful for the reader if a supplementary figure could be made that contains side-by-side structures of S1P/Spring (e.g Fig 3a) and S1P with A domain (e.g Fig 5a) in the exact same orientation so that it is easier to visualize

how SPRING and the A domain have overlapping binding sites. I found myself flipping back and forth between Figure 3 and Figure 5 to compare them.

We thank the reviewer for this suggestion on how to improve the presentation of the structural results. We have incorporated this suggestion to the revised **Fig. 6f-h**, where we now show the S1P structures side by side (**Fig. f**), the SPRING and A domains superposed to show they would clash (**Fig. g**), and their overlapping binding surfaces on S1P colored (**Fig. h**). We hope these revisions clarify this important point.

REVIEWERS' COMMENTS

Reviewer #1 (Remarks to the Author):

Authors have done an excellent job to address all concerns that other reviewers and I have raised on the previous version I only have two minor comments.

1. Line 39 "Additionally, S1P is exploited by Arena viruses to mature their vital glycoproteins and Hepatitis C virus (HCV), which requires SREBP activity." is confusing to me. I would recommend rewording this sentence. For example, S1P is exploited by Arena viruses to mature their vital glycoproteins and by Hepatitis C virus where S1P and SREBP are required to regulate viral lifecycle.

2. Rebuttal letter, the response to reviewer 1 #3, authors are referring supplemental Fig. 3k, not 3g, which is a typo. Authors should double check the document to make sure that all figures and supplemental figures are cited correctly.

Reviewer #2 (Remarks to the Author):

The authors have adequately addressed all my concerns

REVIEWERS' COMMENTS

Reviewer #1 (Remarks to the Author):

Authors have done an excellent job to address all concerns that other reviewers and I have raised on the previous version I only have two minor comments.

We thank the reviewer for their positive assessment of our work and for their suggestions, which have strengthened the manuscript.

1. Line 39 "Additionally, S1P is exploited by Arena viruses to mature their vital glycoproteins and Hepatitis C virus (HCV), which requires SREBP activity." is confusing to me. I would recommend rewording this sentence. For example, S1P is exploited by Arena viruses to mature their vital glycoproteins and by Hepatitis C virus where S1P and SREBP are required to regulate viral lifecycle.

We have revised this sentence as suggested by the reviewer. The revised sentence is: Additionally, S1P is exploited by Arena viruses to mature their viral glycoproteins⁹⁻¹¹ and by Hepatitis C virus (HCV), where S1P and SREBP are required to regulate the viral lifecycle¹².

2. Rebuttal letter, the response to reviewer 1 #3, authors are referring supplemental Fig. 3k, not 3g, which is a typo. Authors should double check the document to make sure that all figures and supplemental figures are cited correctly.

We thank the reviewer for catching this typo and have carefully reviewed the manuscript for errors in the figure citations.

Reviewer #2 (Remarks to the Author):

The authors have adequately addressed all my concerns

We thank the reviewer for their positive assessment of our work and for their suggestions, which have strengthened the manuscript.